# Deep Thinking on Out-Of-Distribution Data: How can we know when a model is overthinking?

## Abstract

Deep thinking models, a class of recurrent architectures, can generalize from easy to hard examples by allocating more computation during inference. While effective in logical reasoning tasks, their potential for test-time adaptation in computer vision under out-of-distribution (OOD) data remains underexplored. This work investigates deep thinking as a test-time scaling strategy for object recognition under distributed shift settings. We show that while thinking longer can improve performance, it also introduces the risk of *overthinking*, where excessive computation damages accuracy. To address this, we propose a self-supervised proxy task that dynamically detects overthinking and approximates the peak accuracy without requiring ground-truth labels. Across multiple OOD object-recognition benchmarks, deep thinking with our proxy delivers performance gains and accuracy close to peak while avoiding overthinking-related drops.

## 1 Introduction

Recurrent neural networks (RNNs), such as Neural Turing Machines (Graves et al., 2014) and Neural GPUs (Kaiser & Sutskever, 2016), have demonstrated a strong extrapolation capability in logical reasoning tasks such as binary addition and multiplication. Deep Thinking (Schwarzschild et al., 2021; Bansal et al., 2022; Veerabadran et al., 2023), a class of recurrent models, further extends these abilities by performing iterative reasoning—a process we refer to as the **Thinking Process**— allowing models trained in simpler problems to solve harder problems by "thinking longer" at inference time. This approach suggests that increasing iterative computation can enhance model generalizability in ways that merely scaling model complexity may not achieve (Zhang et al., 2021; 2024; Prato et al., 2022; Harun et al., 2024).

Generalizing to out-of-distribution (OOD) scenarios is typically a challenging task. Recent work has shown that reasoning-based models, like object-centric networks, can adapt to OOD data by reasoning about objects in varying contexts (Puebla & Bowers, 2024). This mirrors human cognition, where the *Thinking Process* is not fixed but flexibly deepened: we often deliberate longer when inputs are noisy, ambiguous, or unfamiliar (Bar, 2003). Motivated by this analogy, we explore whether deep thinking models can replicate such human-like adaptability, and in particular, whether allowing them to "think longer" at inference helps recognize objects in noisy or blurry images without explicit training on such data.

However, our study reveals that while deep thinking models initially improve with more iterations, excessive reasoning can cause performance to decline, a phenomenon we call *overthinking*. Although this issue has been noted in logical reasoning (Bansal et al., 2022), it is not unique to recurrent models. (Kaya et al., 2018) report similar drops in deep feed-forward networks when depth increases. As shown in Fig. 1, accuracy improves with more steps until an optimal point, after which it degrades. This raises a fundamental question: *if "thinking longer" improves generalization, how long is long enough?*.

Prior work (Graves, 2016; Veerabadran et al., 2023; Mathur et al., 2025) attempted to answer this by halting the Thinking Process with two main strategies: (i) *Adaptive Computation Time* (ACT), which penalizes longer computation via a "ponder cost" and learns to halt; (ii) a norm-threshold heuristic that stops once hidden states appear to converge. While both methods avoid unnecessary

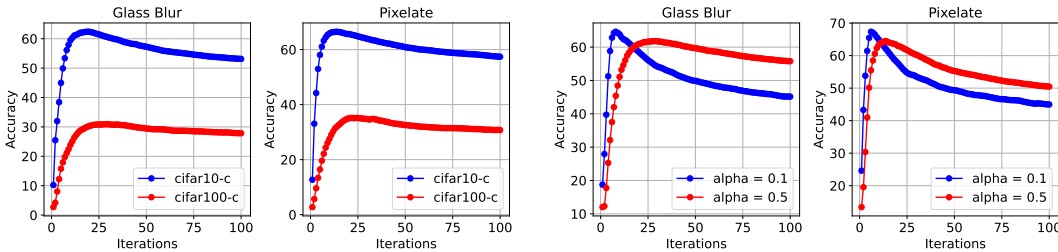

Figure 1: Overthinking occurs on OOD datasets, even when the Deep Thinking Model (Conv-GRU) is trained without (left) and with (right) Progressive Loss.

iterations, they often induce *underthinking*: ACT halts prematurely if the ponder penalty dominates, and the threshold heuristic not only requires hidden states to converge, but also is highly sensitive, with large thresholds making deep thinking models stop too early. Thus, to avoid overthinking, these approaches can inadvertently suppress the depth of the model's reasoning and weaken the extrapolation, especially under OOD data, where longer inference is beneficial.

This motivates a different perspective: instead of strictly halting the Thinking Process, we ask— *if overthinking cannot be fully avoided, how can we detect its onset under test-time distribution shifts without labeled data and select the iteration closest to the accuracy peak?* To this end, we propose using a self-supervised auxiliary task ($\mathcal{T}_a$) as a proxy to monitor the Thinking Process. We hypothesize that $\mathcal{T}_a$ mirrors the main task ($\mathcal{T}_m$): its accuracy rises, peaks, and then falls as overthinking sets in. Since $\mathcal{T}_a$ always has known labels, it provides a practical signal to estimate the optimal reasoning depth without supervision at test time.

Our contributions are threefold:

- We explore the extrapolation capabilities of deep thinking architectures for object recognition. We show that extending the Thinking Process enables strong generalization in OOD settings as a form of test-time scaling.
- We introduce a novel self-supervised approach to detect the onset of *overthinking* and adaptively select the optimal reasoning depth, moving beyond strict halting rules that often induce *underthinking* and limit extrapolation.
- Through experiments across diverse OOD benchmarks, we validate that our approach consistently achieves near-peak accuracy, generalizes robustly across tasks, and requires no additional supervision or few-shot training.

## 2 RELATED WORK

### 2.1 DEEP THINKING AND OVERTHINKING

Deep thinking models are recurrent architectures that generalize from easy to hard examples by repeating layers multiple times at the test time (Schwarzschild et al., 2021). This approach enables extrapolation without training on harder samples. However, (Bansal et al., 2022) identified a critical limitation: *overthinking*, where model accuracy degrades after a certain number of iterations, even after reaching peak performance. They proposed two mitigation strategies: (1) *Recall*, which adds residual connections from the input to each recurrent step, and (2) *Progressive Loss*, which encourages hidden states to converge by training on intermediate representations. (Linsley et al., 2020) applied Lipschitz constraints to ensure convergence in recurrent convolution neural nets, while DEQ (Bai et al., 2019) used fixed-point iterations to define infinitely deep architectures. Despite these efforts, we find that overthinking can still occur even when hidden states converge, suggesting the problem is more complex than simply enforcing stability.

### 2.2 ADAPTIVE COMPUTATION TIME

The idea of adapting inference depth to input difficulty has been formalized by Adaptive Computation Time (ACT) (Graves, 2016), which allows recurrent models to dynamically halt based on learned halting probabilities. ACT has been applied to vision reasoning tasks like Pathfinder and Maze (Veerabadran et al., 2023; Linsley et al., 2018), and to VQA tasks (Eyzaguirre & Soto, 2020). Ponder-Net (Banino et al., 2021) extended ACT with a learned prior over halting, improving stability during training. Another line of work leverages fixed-point iterations, terminating when the difference between successive hidden states is below a threshold (Mathur et al., 2025). These methods try to minimize the reasoning iterations; however, overemphasis on the "ponder cost" may result in early stopping and underthinking. In contrast, our work avoids strict architectural halting constraints and instead proposes a flexible, label-free self-supervised proxy to detect the iteration that achieves accuracy closest to the peak during test time.

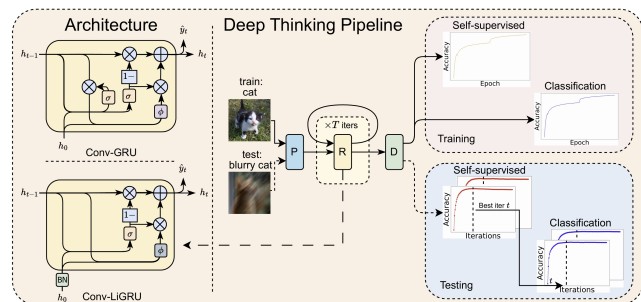

Figure 2: Deep thinking pipeline with training includes both main (classification) and auxiliary (self-supervised) tasks. During inference, the iteration with peak self-supervised performance is selected to estimate the peak accuracy of the classification task. The reset gate in Conv-GRU is simplified, and $\phi$ is replaced from Tanh to ReLU, resulting in Conv-LiGRU.

### 2.3 TEST-TIME TRAINING (TTT)

TTT methods update the model parameters on the test samples without labels, helping the models adapt to distribution shifts (Sun et al., 2020; Gandelsman et al., 2022). Recent work shows that simple TTT can outperform large pretrained models on mathematical reasoning (Muennighoff et al., 2025), and that attention-based updates can improve sequence modeling at test time (Sun et al., 2024). Inspired by this, we propose a self-supervised auxiliary task to estimate the model's optimal reasoning depth at inference. Unlike prior TTT work focused on parameter adaptation, our method uses a frozen model and self-supervision to observe and detect the onset of overthinking without any test labels.

## 3 METHODOLOGY

### 3.1 DEEP THINKING MODEL FOR TEST-TIME SCALING IN OOD TASKS

In this section, we outline the OOD setting and our framework for visual reasoning. We adopt *Deep Thinking* as the baseline, which leverages iterative computation to scale and generalize to unseen data.

**Problem Setting:** Let $\mathcal{D}_{\text{train}}$ and $\mathcal{D}_{\text{test}}$ denote the training and testing sets. Inputs lie in $\mathcal{X} \in \mathbb{R}^{C \times H \times W}$ with labels $\mathcal{Y} = \{1, \ldots, K\}$. Training data is labeled, $\mathcal{D}_{\mathcal{X}, \mathcal{Y}}$, while test data is unlabeled, $\mathcal{D}'_{\mathcal{X}}$. We assume a covariate shift (Hendrycks & Dietterich, 2019), i.e., $\mathcal{D}'_{\mathcal{X}} \neq \mathcal{D}_{\mathcal{X}}$ but with the same label space $\mathcal{Y}$.

Prior work has pursued invariance via representation learning (Lu et al., 2021; Wang et al., 2021) or data augmentation (Volpi et al., 2018). In contrast, (Bansal et al., 2022) show that recurrent models provide a scalable alternative. Our Deep Thinking framework (Figure 2) consists of three stages:

**Input Transformation (P):** A convolutional layer maps the image to the initial hidden state:

$$\mathbf{h}_0 = \sigma(\mathbf{W}_{\text{in}} * \mathbf{x} + \mathbf{b}_{\text{in}}).$$

**Thinking Process (R):** Hidden states are iteratively updated by a recurrent function:

$$\mathbf{h}_t = f_{\text{rec}}(\mathbf{h}_{t-1}, \mathbf{h}_0), \quad t = 1, \ldots, T_{\text{train}},$$

where $f_{\text{rec}}$ is instantiated as Conv-GRU (Ballas et al., 2016) or Conv-LiGRU (Ravanelli et al., 2018). Unlike Recall (Bansal et al., 2022), we integrate $\mathbf{h}_{t-1}$ with $\mathbf{h}_0$ instead of concatenating with $\mathbf{x}$,

reducing computation via input downsampling. In inference, $T_{test}$ can be increased to cope with corrupted inputs.

**Output Prediction (D):** At each step $t$, predictions are made as $\hat{\mathbf{y}}_t = f_{\text{out}}(\mathbf{h}_t)$.

We hypothesize that the **Thinking Process** adaptively allocates more iterations to harder samples, improving robustness under corruption and enhancing the extrapolation capacity of recurrent models in OOD scenarios.

### 3.2 SELF-SUPERVISED ITERATION ESTIMATION

Halting strategies such as ACT or norm-threshold rules aim to prevent unnecessary computation but often induce *underthinking*, as discussed in Section 1, and we show that in Section 4.4.1. Instead of constraining the *Thinking Process*, we allow models to run freely and focus on detecting when performance begins to decline. Our key question is: *how can we identify the onset of overthinking under distribution shifts without access to test labels?*

We address this by introducing a self-supervised auxiliary task ($\mathcal{T}_a$) as a proxy. Since $\mathcal{T}_a$ has known labels and its accuracy mirrors the trend of the main task ($\mathcal{T}_m$)—rising to a peak and then dropping—tracking $\mathcal{T}_a$ provides a reliable signal to estimate the optimal reasoning depth.

---

**Algorithm 1** Testing Phase: Accuracy-Iteration Relationship Estimation

---

1: **Input:** Trained model, test data with $D$ batches, maximum iterations $T_{\text{test}}$
2: **Output:** Optimal iteration $t_{\text{opt}}$

3: **Step 1: Initialize variables**
4: Initialize `Correct` as a zero array of length $T_{\text{test}}$

5: **Step 2: Compute Accuracy$_{\mathcal{T}_{\text{aux}}}(t)$ during testing**
6: **for** $i = 1, 2, \ldots, D$ **do**  ▷ Iterate over $D$ test batches
7:     RotatedBatch$_i$, RotatedLabel$_i \leftarrow$ RotateFunction(Batch$_i$)
8:     $h^{(0)} \leftarrow$ InputTransformation(Batch$_i$)
9:     $h^{(0)}_{rotate} \leftarrow$ InputTransformation(RotatedBatch$_i$)
10:    **for** $t = 1, 2, \ldots, T_{\text{test}}$ **do**  ▷ Iterate over $T_{\text{test}}$ steps
11:        $h^{(t)} \leftarrow f_{\text{rec}}(h^{(t-1)}, h^{(0)})$  ▷ Recurrent computation
12:        $h^{(t)}_{rotate} \leftarrow f_{\text{rec}}(h^{(t-1)}_{rotate}, h^{(0)}_{rotate})$
13:        $\hat{y}^{(t)}_{main} \leftarrow f_{\text{out}}(h^{(t)})$  ▷ Output predictions
14:        $\hat{y}^{(t)}_{rotate} \leftarrow f_{\text{rotate}}(h^{(t)}_{rotate})$
15:        `Correct`$[t]$ += # Correct auxiliary samples
16:    **end for**
17: **end for**
18: Accuracy$_{\mathcal{T}_{\text{aux}}}(t) \leftarrow$ `Correct`$[t]$/TotalSamples

19: **Step 3: Estimate optimal iteration**
20: $t_{\text{opt}} \leftarrow \arg\max_t \left(\text{Accuracy}_{\mathcal{T}_{\text{aux}}}(t)\right)$

21: **Return:** $t_{\text{opt}}$

---

**Assumption.** First, let us denote $\hat{y}^{(a)}_t = \arg\max_{k=1:K} \hat{\mathbf{y}}^{(a)}_t[k]$ and $y^{(a)}$ as the prediction and the ground truth labels of the auxiliary task, which is observed. Then, we introduce the essential assumption for our subsequent method.

*(A.1)* The correlation between $\mathcal{T}_a$ and $\mathcal{T}_m$ is positive, i.e., $\forall t, \text{corr}\left(\mathbb{P}\left[\hat{y}^{(a)}_t = y^{(a)}\right], \mathbb{P}\left[\hat{y}^{(m)}_t = y^{(m)}\right]\right) > 0$.

*(A.2)* The correlation exists under both ID and OOD settings.

Following (Balaji et al., 2018), we treat the auxiliary task as a regularization objective and train it jointly with the main task. This encourages the model to learn domain-invariant features that generalize across shifts (Assumption 1). In addition, (Sun et al., 2020) show that optimizing the auxiliary task at test time further adapts the model to the target distribution (Assumption 2).

While various self-supervised tasks (e.g., jigsaw puzzles (Carlucci et al., 2019)) could in principle serve this role, our focus is not on designing new auxiliary objectives but on using them as a proxy to monitor overthinking in OOD scenarios. We adopt rotation prediction (Sun et al., 2020; Richardson & Weiss, 2020) as $\mathcal{T}_a$ due to its simplicity, ease of implementation, and demonstrated effectiveness in satisfying our assumptions. Specifically, each input $\mathbf{x}$ is randomly rotated by one of $\{0°, 90°, 180°, 270°\}$, and the model is trained to predict the rotation angle. The auxiliary loss at iteration $t$ is given by:

$$\mathcal{L}^{(a)}_t = -\sum_{k=1}^{4} \mathbf{y}^{(a)}[k] \log \hat{\mathbf{y}}^{(a)}_t[k],$$

and the total training objective is the sum of the main and auxiliary losses at the final iteration:

$$\mathcal{L} = \mathcal{L}^{(m)}_{T_{\text{train}}} + \mathcal{L}^{(a)}_{T_{\text{train}}}.$$

Table 1: The number of parameters and the peak accuracy (%) over all 15 types of corruption at level 5 in the CIFAR10-C and CIFAR100-C datasets

| Model | Params | CIFAR10-C | CIFAR100-C |
|---|---|---|---|
| ViT | 9.5M | 51.7 | 26.8 |
| Resnet | 18.4M | 60.7 | 22.6 |
| Conv-LiGRU | 1.3M | 62.6 | 27.9 |
| Conv-GRU | 1.6M | **67.1** | **31.9** |

**Iteration Search:** At test time, we find the iteration with the highest auxiliary accuracy:

$$t_{\text{opt}} = \arg \max_{t \in [T_{\text{test}}]} \mathbb{P}\left(\hat{y}_t^{(a)} = y^{(a)}\right), \tag{1}$$

then predict the main task with $\hat{y}_{t_{\text{opt}}}^{(m)}$. The complete procedure is detailed in the algorithm 1.

## 4 EXPERIMENT

### 4.1 DATASETS

We evaluate the extrapolation ability of deep thinking models using both standard and corrupted datasets. Models are trained separately on CIFAR10, CIFAR100 (Alex Krizhevsky, 2009), and Tiny ImageNet (Ya Le, 2015), and tested on their corrupted variants: CIFAR10-C, CIFAR100-C, and Tiny ImageNet-C (Hendrycks & Dietterich, 2019) with 15 corruption types at 5 severity levels. To investigate the extrapolation capacity on the unseen distribution, we use STL10 (Coates et al., 2011) ($96 \times 96$), which shares nine classes with CIFAR10 ($32 \times 32$) plus an outlier. Models are trained only on CIFAR10 and evaluated directly on STL10 without resizing the input scale.

Figure 3: **Conv-GRU vs. Conv-LiGRU.** Left axis: step norm $\|\Delta h_t\|$; right axis: accuracy. Conv-LiGRU has tiny, rapidly shrinking steps and shows no overthinking; Conv-GRU has larger steps, slower decay, and gradually drops post-peak accuracy.

### 4.2 MODELS AND TRAINING

For recurrent networks, we employed two recurrent CNN variants: Conv-GRU and Conv-LiGRU. These architectures allow iterative reasoning through shared convolutional recurrent layers. During training, we used a fixed number of iterations $T_{\text{train}} = 30$. We adopt ResNet-30 without batch normalization as the backbone, following prior work (Schwarzschild et al., 2021; Veerabadran et al., 2023). It consists of 30 residual blocks with two convolutional layers each, mimicking the depth and capacity of recurrent models while using unshared weights across layers. We also compare extrapolation performance with Vision Transformer (ViT) (Dosovitskiy et al., 2021), adapted for CIFAR10 and CIFAR100 (Yoshioka, 2024).

All models are trained with self-supervised learning, each input image was randomly rotated using a function $f_{rotate}$ selecting from $\{0°, 90°, 180°, 270°\}$. The details of batch sizes, learning rate, decay schedules, and other hyperparameters can be found in A.6.

### 4.3 DEEP THINKING EXTRAPOLATION CAPABILITY

#### 4.3.1 ROBUSTNESS AND ADAPTABILITY.

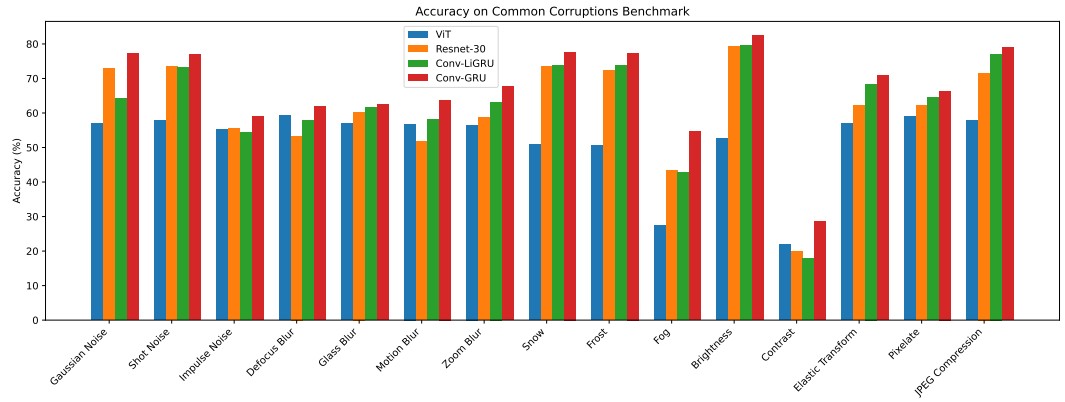

Figure 4: Peak accuracy (%) on CIFAR10-C, at level 5, ViT, Resnet-30, Conv-LiGRU and Conv-GRU

Table 2: Peak accuracy (%) on Tiny ImageNet-C, level 5, Resnet-30, Conv-GRU

|        | gauss | shot | impul | defoc | glass | motn | zoom | snow | frost | fog | brit | contr | elast | pixel | jpeg | Avg. |
|--------|-------|------|-------|-------|-------|------|------|------|-------|-----|------|-------|-------|-------|------|------|
| resnet | 9.9   | 11.8 | 7.2   | **3.1** | **3.0** | **7.8** | **7.4** | 13.1 | 12.7 | **5.4** | **15.6** | 1.0 | 10.5 | 16.3 | 16.7 | 9.4 |
| c-gru  | **13.0** | **14.4** | **9.6** | 2.0 | 2.5 | 6.6 | 5.4 | **15.8** | **15.5** | 5.2 | 13.6 | **1.6** | **10.8** | **16.4** | **18.8** | **10.1** |

Deep thinking with recurrent models is well known for its generalization in logical reasoning, and we extend this property to computer vision. Unlike feed-forward networks with fixed computation, recurrent models adaptively allocate more iterations when inputs are heavily corrupted. Figure 5 illustrates this: Conv-GRU "thinks longer" at corruption level 5 than at level 3, yet still delivers stable accuracy gains even without exposure to such severe noise during training. In contrast, ResNet exhibits slower and less stable improvements across layers, with accuracy fluctuating rather than converging smoothly. Conv-GRU secures most of its gains within 10–15 iterations and continues refining predictions up to 30, highlighting the stability and robustness of iterative reasoning. In particular, with 30 iterations, Conv-GRU sur-

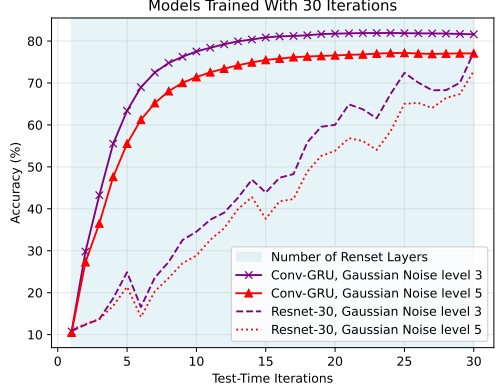

Figure 5: Evaluation of the network's ability to classify objects on test sets with higher noise severity than training.

passes ResNet in equivalent depth, demonstrating the strong extrapolation capability of deep thinking models.

### 4.3.2 EFFICIENT AND LIGHTWEIGHT

Figure 4 shows that Conv-GRU consistently outperforms Resnet and ViT across nearly all corruption categories. Notably, Conv-GRU demonstrates strong robustness to more challenging corruptions, such as fog and contrast, where others suffer significant performance drops. Table 1 shows the average accuracy across 15 level-5 corruption datasets. Despite having significantly fewer parameters, Conv-LiGRU and Conv-GRU achieve higher peak accuracy. This underscores the efficiency of recurrent deep thinking architectures while generalizing well under severe corruption.

### 4.3.3 SCALABILITY.

Table 2 reports peak accuracy on Tiny ImageNet-C (level 5). Conv-GRU consistently outperforms Resnet across most corruption types. Notable gains are observed in noise corruptions (e.g., +3.1%

on Gaussian, +2.6% on Shot) and spatial distortions like snow, frost, and JPEG compression. These improvements highlight Conv-GRU's ability to adapt to high-resolution contexts where fixed-depth models struggle. Although ResNet slightly outperforms Conv-GRU on a few blur-related corruptions, the overall trend demonstrates that deep thinking models maintain generalization even on higher-resolution datasets. In A.5, we further visualize how deep thinking models extract features during inference by examining the patterns the model detects at each "thinking step". We provide detailed experimental results in A.8.

## 4.4 DETECTING OVERTHINKING WITH SELF-SUPERVISION

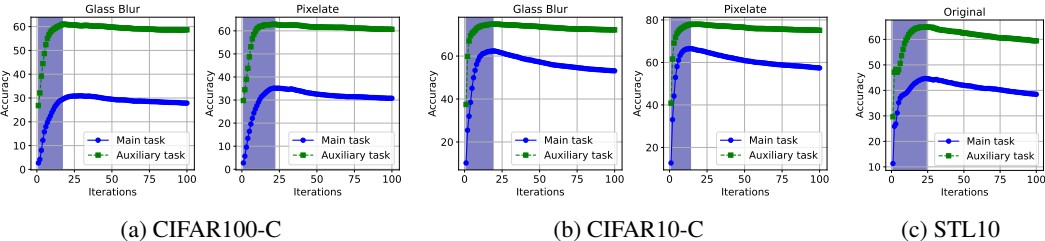

| (a) CIFAR100-C | (b) CIFAR10-C | (c) STL10 |

Figure 6: Correlation between the accuracy of the main and auxiliary tasks for Conv-GRU on OOD datasets. The dark blue region indicates the recurrent steps prior to the onset of overthinking, as estimated by our proposed algorithm (1).

As shown in Figure 1, when $T_{\text{test}} = 100$, the problem of overthinking becomes evident. The accuracy declines sharply after peak accuracy, and overthinking occurs across all 15 corruption test sets (see A.9). To address overthinking, (Bansal et al., 2022) proposed Progressive Loss, aiming to ensure that the iterative process converges consistently from any starting point and reducing the risk of overthinking. However, Figure 1 also shows that overthinking remains prevalent even after applying Progressive Loss with different coefficients $\alpha = 0.1$ and $\alpha = 0.5$. The accuracy drops significantly after reaching its peak, indicating that this approach alone cannot fully resolve overthinking. The details of the Progressive Loss algorithm are described in A.4.

### 4.4.1 ANALYSIS OF THE UNDERTHINKING PROBLEM

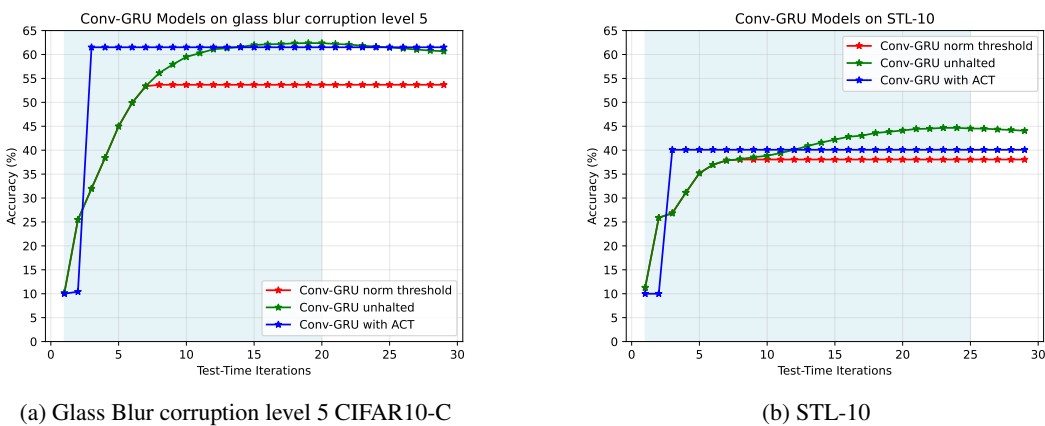

| (a) Glass Blur corruption level 5 CIFAR10-C | (b) STL-10 |

Figure 7: Accuracy of Conv-GRU under three settings on Glass Blur level 5 from CIFAR10-C (*near OOD*, $32 \times 32$) and STL-10 (unseen OOD, $96 \times 96$). The shaded blue region marks the recurrent steps before the onset of overthinking, estimated by our algorithm (Algorithm 1). Note that Conv-GRU is trained only on CIFAR-10 and evaluated directly on STL-10 without input resizing.

We compare Conv-GRU under three contrasting setups. (1) **Unhalted**: inference proceeds without constraints, and we use our self-supervised estimator (Algorithm 1) to approximate the peak accuracy. (2) **Norm-threshold rule**: inference halts once $\|\mathbf{h}_t - \mathbf{h}_{t-1}\| / \|\mathbf{h}_t\| < \varepsilon$, with $\varepsilon = 0.03$ in our

Table 3: Accuracy and iteration statistics (mean and standard deviation) for estimating peak accuracy on extrapolation datasets. Note: ResNet has a fixed depth (30 layers), hence avg = 30 and std = 0. We set the threshold $\varepsilon = 0.03$ for the norm threshold rule.

| Model | Cifar10C % | STL10 % | Avg $t_{opt}$ | Std $t_{opt}$ |
|---|---|---|---|---|
| Resnet | 60.7 | 24.6 | 30 | 0 |
| Conv-GRU (ACT) | 62.76 | 40.08 | 3.0 | 0.0 |
| Conv-GRU (threshold rule $\varepsilon$) | 60.25 | 38.04 | 7.4 | 0.5 |
| Conv-GRU (our algorithm 1) | **66.1** | **44.5** | 24.6 | 9.6 |
| Conv-GRU (peak) | 67.1 | 44.7 | 20.1 | 4.4 |

experiments. Prior work (Veerabadran et al., 2023) reported that $\varepsilon = 0.1$ leads to halting after only two steps; we adopt a smaller threshold to allow more reasoning iterations. (3) **ACT**: the Adaptive Computation Time mechanism is applied during training to jointly optimize the halting policy and inference process. We provide a more detailed explanation of ACT in A.2. To ensure fairness, the self-supervised task is included in all setups.

Figure 7 presents the accuracy across iterations in these settings. The norm threshold rule consistently stops too early (red line), both on *near*-OOD (CIFAR-10C) and unseen OOD data (STL-10), highlighting its sensitivity to convergence dynamics. ACT also halts quickly, after only $\sim$3 iterations. Although this is sufficient for CIFAR-10C, where ACT performance matches the unhalted baseline, it becomes a clear bottleneck on STL-10, where the unhalted model surpasses ACT in $\sim$ 13 iterations and continues to improve until a peak near $\sim$24. This contrast underscores the limitations of ACT in handling tougher distribution shifts that require deeper reasoning.

### 4.4.2 EFFICIENCY OF SELF-SUPERVISED ITERATION ESTIMATION

Rather than constraining the *Thinking Process* with fixed halting rules, we allow deep thinking models to run freely and use a self-supervised auxiliary task as a proxy to monitor accuracy trends. Because this task is label-free yet closely tracks the main objective, it provides a reliable signal for estimating the optimal iteration, even under challenging shifts such as STL-10 with unseen $96 \times 96$ inputs (Figure 6).

Table 3 validates our approach. ACT halts deterministically after only three iterations—adequate for mild shifts but a severe bottleneck on STL-10. The norm-threshold rule allows more iterations, yet without learning to optimize the reasoning process it remains less effective. In contrast, our method adaptively estimates reasoning depth without labels or thresholds, achieving accuracies (66.1% / 44.5%) nearly identical to the oracle peaks. Although $t_{opt}$ may exceed the oracle, the slow post-peak decay makes this negligible. Thus, the auxiliary signal strikes a balance, preventing both premature halting and overthinking, and enabling robust extrapolation.

Moreover, Table 3 shows that Conv-GRU with our estimation consistently outperforms ResNet despite the scale mismatch. Even Conv-GRU with ACT, limited to three iterations, surpasses ResNet—highlighting that deep thinking models inherently possess strong extrapolation ability, not merely an adjusted receptive field. Our estimation further preserves and amplifies this advantage, allowing deep thinking models to generalize effectively beyond training resolution. We provide a detail analysis of self-supervised iteration estimation in the Appendix A.7.

## 5 DISCUSSION: THE CONTRACTION ASSUMPTION AND OVERTHINKING

In this section, we want to outline a plausible intuition, the *contraction assumption*, which can help to provide another view into overthinking. (Bansal et al., 2022) suggested that divergence of hidden states $\mathbf{h}_t$ may drive overthinking. Yet Figure 3 shows this is not the whole story: Conv-GRU suffers accuracy drops even when $\|\mathbf{h}_t - \mathbf{h}_{t-1}\| \to 0$. Thus, overthinking is subtler than non-convergence—merely forcing hidden states to a fixed point is insufficient. Motivated by this, we analyze a simplified recurrent model under convex and smooth loss assumptions to illustrate the intuition behind the contraction assumption before turning to our empirical findings.

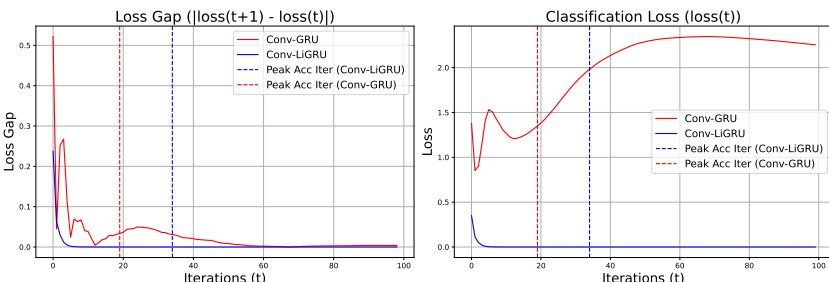

Figure 8: Loss gap and classification loss for Conv-GRU and Conv-LiGRU.

**Lemma 1** (Bounded Drift under Contraction). *Consider the linear recurrent model at test time*

$$\mathbf{h}_{t+1} = A\mathbf{h}_t + B\mathbf{h}_0, \quad \Delta\mathbf{h}_t = \mathbf{h}_{t+1} - \mathbf{h}_t.$$

*Let the task loss $\mathcal{L}_m(\mathbf{h})$ be $L$-smooth with bounded gradient $\|\nabla\mathcal{L}_m(\mathbf{h}_t)\| \leq G$. Then the per-step loss drift is bounded by*

$$|\mathcal{L}_m(\mathbf{h}_{t+1}) - \mathcal{L}_m(\mathbf{h}_t)| \leq G\|\Delta\mathbf{h}_t\| + \tfrac{L}{2}\|\Delta\mathbf{h}_t\|^2.$$

*Moreover, if the updates contract geometrically, i.e. $\|\Delta\mathbf{h}_{t+1}\| \leq c\|\Delta\mathbf{h}_t\|$ for some $c \in (0,1)$, then the cumulative drift after step $t$ is uniformly bounded:*

$$\sum_{k\geq 0} |\mathcal{L}_m(\mathbf{h}_{t+k+1}) - \mathcal{L}_m(\mathbf{h}_{t+k})| \leq \tfrac{G}{1-c}\|\Delta\mathbf{h}_t\| + \tfrac{L}{2(1-c^2)}\|\Delta\mathbf{h}_t\|^2.$$

The proof is provided in Appendix A.3. To avoid overthinking, we assume that the model's prediction values after the peak iteration should remain stable, or equivalently, that the loss gap between the peak iteration and later iterations must stay small. Lemma 1 shows that this gap is upper-bounded by a monotonically increasing quadratic function of $\|\Delta\mathbf{h}_t\|$. Therefore, as $\|\Delta\mathbf{h}_t\|$ decreases, this bounding function also decreases, leading to a smaller loss gap. However, if $\|\Delta\mathbf{h}_t\|$ remains large at the time the model reaches peak accuracy, the loss gap may shrink slowly, causing the model's prediction values in post-peak iterations to change and ultimately harming accuracy. This theoretical prediction aligns with our empirical observations. In Figure 3, the $\|\Delta\mathbf{h}_t\|$ of both Conv-GRU and Conv-LiGRU eventually converges to $0$, yet only Conv-GRU suffers from overthinking, while the accuracy of Conv-LiGRU remains unaffected. Figure 8 further shows that the loss gap of both models decreases as $\|\Delta\mathbf{h}_t\|$ decreases, consistent with Lemma 1. However, at the peak iteration, the $\|\Delta\mathbf{h}_t\| \approx 0.71$ of Conv-GRU is still large, causing the loss gap to converge slowly and leading to post-peak loss divergence. In contrast, at the peak iteration of Conv-LiGRU, $\|\Delta\mathbf{h}_t\| \approx 0.004$ is already small and close to $0$, which makes the loss gap converge quickly and prevents its accuracy from being harmed. So, from the lemma 1 and the empirical findings we discuss above, we consider that besides $h_t$ converging condition, the absolute scale of $\|\Delta\mathbf{h}_t\|$ jointly determines whether the additional reasoning remains stable or harmful.

## 6 CONCLUSION

This paper studied the extrapolation ability of deep thinking architectures in object recognition, showing that extending the *Thinking Process* serves as an effective form of test-time scaling for OOD generalization. We found that heuristic halting rules, though designed to reduce computation, often induce *underthinking*. To address this, we proposed a lightweight self-supervised auxiliary task that tracks accuracy trends and identifies near-peak performance without labels. Experiments across diverse OOD benchmarks confirm that our method achieves near-optimal accuracy and robust generalization without additional supervision. Overthinking, however, remains an open challenge. Our estimator is a first step, but deeper theoretical and empirical understanding is needed of why and when overthinking arises. We hope this work encourages broader use of self-supervised tasks as monitoring signals for main-task performance, offering a domain-agnostic way to adapt iterative models at test time. Future work will explore extending our framework to logical reasoning and large language models, where iterative computation is central and the tension between underthinking and overthinking may be even sharper.

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

# A APPENDIX

## A.1 DETAILS OF RECURRENT MODULES: CONV-GRU AND CONV-LIGRU

**Conv-GRU** is a convolutional variant of the Gated Recurrent Unit (GRU), designed to handle spatial data such as images. Instead of fully connected transformations, Conv-GRU replaces them with convolutional operations, allowing spatial information to be preserved throughout the recurrence. At each time step $t$, the hidden state $\mathbf{h}_t$ is updated as:

$$
\begin{aligned}
\mathbf{z}_t &= \sigma(\mathbf{W}_z * \mathbf{h}_{t-1} + \mathbf{U}_z * \mathbf{h}_0), \\
\mathbf{r}_t &= \sigma(\mathbf{W}_r * \mathbf{h}_{t-1} + \mathbf{U}_r * \mathbf{h}_0), \\
\tilde{\mathbf{h}}_t &= \tanh(\mathbf{W}_h * (\mathbf{r}_t \odot \mathbf{h}_{t-1}) + \mathbf{U}_h * \mathbf{h}_0), \\
\mathbf{h}_t &= (1 - \mathbf{z}_t) \odot \mathbf{h}_{t-1} + \mathbf{z}_t \odot \tilde{\mathbf{h}}_t,
\end{aligned}
$$

where $*$ denotes convolution, $\odot$ is the element-wise product, and $\sigma(\cdot)$ is the sigmoid function.

**Conv-LiGRU** (Lightweight Conv-GRU) simplifies the Conv-GRU by removing the reset gate and replacing the tanh activation with ReLU. This results in a more efficient model with fewer parameters and faster training while still maintaining the core recurrence mechanism. The update rule becomes:

$$
\begin{aligned}
\mathbf{z}_t &= \sigma(\mathbf{W}_z * \mathbf{h}_{t-1} + \mathbf{U}_z * \mathbf{h}_0), \\
\tilde{\mathbf{h}}_t &= \text{ReLU}(\mathbf{W}_h * \mathbf{h}_{t-1} + \mathbf{U}_h * \mathbf{h}_0), \\
\mathbf{h}_t &= (1 - \mathbf{z}_t) \odot \mathbf{h}_{t-1} + \mathbf{z}_t \odot \tilde{\mathbf{h}}_t.
\end{aligned}
$$

Both variants enable efficient iterative reasoning over spatial inputs and are used as the core recurrent modules in our Deep Thinking framework.

## A.2 BACKGROUND: ADAPTIVE COMPUTATION TIME (ACT)

ACT Graves (2016) is a mechanism designed to dynamically determine the number of recurrent steps required to process each input. Unlike its original formulation, which handles variable-length sequences, our work applies ACT to static inputs in visual reasoning tasks.

At each time step, the model generates a halting score $p_t$ through a learned convolutional layer. The cumulative sum of these scores, $P_t$, determines whether the computation should continue. When $P_t$ reaches a predefined threshold $(1 - \epsilon)$, iteration stops, and the final hidden state is computed as a weighted sum of the intermediate states.

A key component of ACT is the ponder cost, an auxiliary loss term that encourages the model to minimize the number of recurrent steps while maintaining accuracy. The total loss function consists of the task loss $L_{task}(y, \hat{y}_{act})$ and the ponder cost, weighted by a hyperparameter $\tau$ (Equal 2). In our study, we set $\tau = 0.5$ to analyze the limitations of ACT's early stopping heuristic.

$$
\mathcal{L} = \sum_{i=0}^{|\mathcal{D}|} \frac{1}{|\mathcal{D}|} L_{task}(y^i, \hat{y}_{\text{act}}^i) - \tau \sum_{t=1}^{t_{\text{halt}}^i - 1} p_t^i \tag{2}
$$

## A.3 LEMMA (TAIL DRIFT UNDER $L$-SMOOTHNESS AND GEOMETRIC CONTRACTION).

Assume the main loss $\mathcal{L}_m : \mathbb{R}^d \to \mathbb{R}$ is $L$-smooth and its gradient is bounded along the test-time trajectory: $\|\nabla \mathcal{L}_m(\mathbf{h}_t)\| \le G$ for all $t$. Let $\Delta \mathbf{h}_t := \mathbf{h}_{t+1} - \mathbf{h}_t$ and suppose there exists $c \in (0, 1)$ such that $\|\Delta \mathbf{h}_{t+1}\| \le c \|\Delta \mathbf{h}_t\|$ for all $t$. Then

$$
\sum_{k=0}^{\infty} \left| \mathcal{L}_m(\mathbf{h}_{t+k+1}) - \mathcal{L}_m(\mathbf{h}_{t+k}) \right| \le \frac{G}{1-c} \|\Delta \mathbf{h}_t\| + \frac{L}{2(1-c^2)} \|\Delta \mathbf{h}_t\|^2.
$$

**Proof.** **(1) Two-sided Descent Lemma.** Since $\mathcal{L}_m$ is $L$-smooth, for all $\mathbf{x}, \mathbf{y}$,

$$
\left| \mathcal{L}_m(\mathbf{y}) - \mathcal{L}_m(\mathbf{x}) - \langle \nabla \mathcal{L}_m(\mathbf{x}), \mathbf{y} - \mathbf{x} \rangle \right| \le \frac{L}{2} \|\mathbf{y} - \mathbf{x}\|^2. \tag{DL}
$$

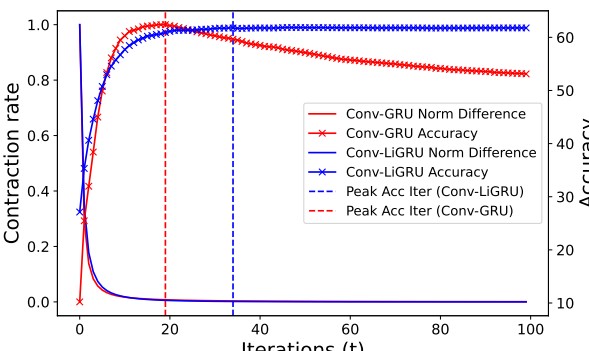

Figure 9: Contraction rates $c = \|\mathbf{h}_t - \mathbf{h}_{t-1}\|/\|\mathbf{h}_t\|$ and accuracies of Conv-GRU and Conv-LiGRU across reasoning iterations.

Set $\mathbf{x} = \mathbf{h}_{t+k}$, $\mathbf{y} = \mathbf{h}_{t+k+1}$, and write $\Delta\mathbf{h}_{t+k} = \mathbf{y} - \mathbf{x}$. Then

$$
\begin{aligned}
\left|\mathcal{L}_m(\mathbf{h}_{t+k+1}) - \mathcal{L}_m(\mathbf{h}_{t+k})\right| &\leq \left|\langle\nabla\mathcal{L}_m(\mathbf{h}_{t+k}), \Delta\mathbf{h}_{t+k}\rangle\right| + \tfrac{L}{2}\|\Delta\mathbf{h}_{t+k}\|^2 \\
&\leq \|\nabla\mathcal{L}_m(\mathbf{h}_{t+k})\|\|\Delta\mathbf{h}_{t+k}\| + \tfrac{L}{2}\|\Delta\mathbf{h}_{t+k}\|^2 \\
&\leq G\|\Delta\mathbf{h}_{t+k}\| + \tfrac{L}{2}\|\Delta\mathbf{h}_{t+k}\|^2.
\end{aligned}
\tag{1}
$$

**(2) Apply geometric contraction to the steps.** By hypothesis, $\|\Delta\mathbf{h}_{t+k}\| \leq c^k\|\Delta\mathbf{h}_t\|$ with $c \in (0,1)$. Summing (1) over $k = 0, 1, 2, \dots$ gives

$$
\begin{aligned}
\sum_{k=0}^{\infty}\left|\mathcal{L}_m(\mathbf{h}_{t+k+1}) - \mathcal{L}_m(\mathbf{h}_{t+k})\right| &\leq G\sum_{k=0}^{\infty}\|\Delta\mathbf{h}_{t+k}\| + \frac{L}{2}\sum_{k=0}^{\infty}\|\Delta\mathbf{h}_{t+k}\|^2 \\
&\leq G\sum_{k=0}^{\infty}c^k\|\Delta\mathbf{h}_t\| + \frac{L}{2}\sum_{k=0}^{\infty}c^{2k}\|\Delta\mathbf{h}_t\|^2 \\
&= \frac{G}{1-c}\|\Delta\mathbf{h}_t\| + \frac{L}{2(1-c^2)}\|\Delta\mathbf{h}_t\|^2,
\end{aligned}
$$

because the geometric series satisfy $\sum_{k\geq 0}c^k = \frac{1}{1-c}$ and $\sum_{k\geq 0}c^{2k} = \frac{1}{1-c^2}$. $\qquad\square$

**Immediate corollary (one-step drift).** Taking $k = 0$ in (1) yields

$$
\left|\mathcal{L}_m(\mathbf{h}_{t+1}) - \mathcal{L}_m(\mathbf{h}_t)\right| \leq G\|\Delta\mathbf{h}_t\| + \tfrac{L}{2}\|\Delta\mathbf{h}_t\|^2.
$$

As illustrated in Figure 3, Conv-LiGRU exhibits small, rapidly shrinking updates and quickly stabilizes, whereas Conv-GRU maintains larger, slower-decaying steps and shows a clear post-peak drop. Figure 9 further demonstrates that even under comparable contraction rates, overthinking persists in Conv-GRU but not in Conv-LiGRU. This indicates that both the contraction rate $c$ and the update magnitude $\|\Delta\mathbf{h}_t\|$ jointly determine whether continued reasoning remains stable or leads to accuracy degradation.

### A.4 PROGRESSIVE LOSS

**Progressive Loss:** To mitigate overthinking, Bansal et al. (2022) proposes the Progressive Loss method 2, which encourages an improvement in the quality of prediction in any thinking step. Beyond computing the loss at the final iteration $h_{T_{\text{train}}}$, Progressive Loss selects a random intermediate step $n \in [1, T_{\text{train}}]$, performs $k = \text{random}(1, T_{\text{train}} - n)$ additional iterations, and computes a loss for this partial forward pass. This loss, scaled by a coefficient $\alpha$, is combined with the final-step loss, ensuring the iterative process converges consistently from any starting point and reducing the risk of overthinking.

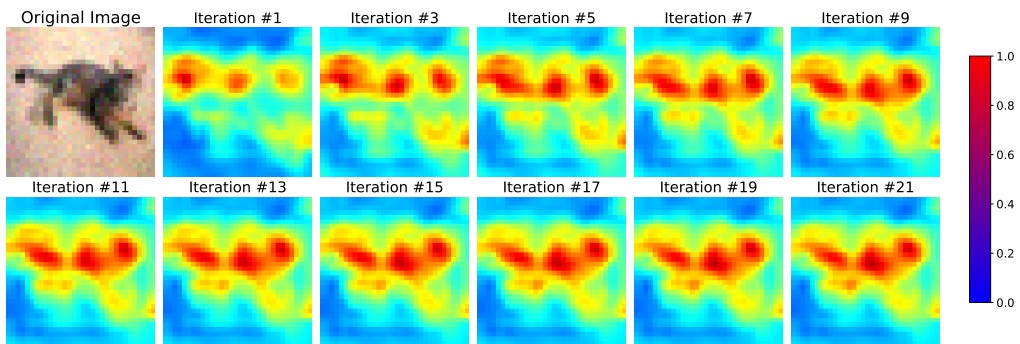

Figure 10: A "Cat" sample input, and outputs from different iterations are shown to illustrate the model's sequential reasoning process on CIFAR10-C (level 1). We visualize the norm of vector feature $h_t^{(i,j)}$ (row $i$, column $j$) of the feature map $h_t$, demonstrating the model's feature extraction over iterations. This is a representative example from a Conv-LiGRU model trained on CIFAR10 with $T_{\text{train}} = 30$.

---

**Algorithm 2** Incremental Progress Training Algorithm

---

1: **Input:** parameter vector $\theta$, integer $m$, weight $\alpha$
2: **for** batch_idx $= 1, 2, \ldots$ **do**
3:     Choose $n \sim U\{0, m-1\}$ and $k \sim U\{1, m-n\}$
4:     Compute $\phi_n$ with $n$ iterations w/o tracking gradients
5:     Compute $\hat{y}_{prog}$ with additional $k$ iterations
6:     Compute $\hat{y}_m$ with new forward pass of $m$ iterations
7:     Compute $\mathcal{L}_{\text{max\_iters}}$ with $\hat{y}_m$ and $\mathcal{L}_{\text{progressive}}$ with $\hat{y}_{\text{prog}}$
8:     Compute $\mathcal{L} = (1 - \alpha) \cdot \mathcal{L}_{\text{max\_iters}} + \alpha \cdot \mathcal{L}_{\text{progressive}}$
9:     Compute $\nabla_\theta \mathcal{L}$ and update $\theta$
10: **end for**

---

A.5 ITERATIVE OUTPUTS

To understand the model's thinking process, we visualize the hidden feature maps $h_t$ at each iteration. Figure 10 presents the Gaussian heatmap of the norm vector feature $h_t^{(i,j)}$ (at row $i$, column $j$) for each feature map $h_t$. This figure reveals two notable insights. First, the model progressively detects features from local to global, gradually capturing the entire object. Second, it prioritizes identifying key distinguishing features, such as the face and tail, before detecting less critical ones, like the legs. This suggests that the model's thinking process can be reasoned about and exhibits a naive but intuitive recognition process similar to human perception.

Figure 11 shows that deep thinking models effectively detect invariant features, such as the frog's head, across different levels of corruption. Even at higher corruption levels, the model can identify these key features, demonstrating its robustness. Despite increased noise or distribution shifts, the model consistently identifies critical features, allowing it to generalize effectively in OOD conditions.

A.6 TRAINING DETAIL

Across all datasets, images are first passed through the Input Transformation module (Section 3.1) and downsampled to a resolution of $16 \times 16 \times d$. For CIFAR10 and CIFAR100, $d = 128$, while for Tiny ImageNet, $d = 256$. On Tiny ImageNet, we limit our evaluation to only Conv-GRU and Resnet-30.

All models were trained for 150 epochs across all datasets. We used the Adam optimizer Diederik P. Kingma (2015) with a weight decay of 0.0002. Each dataset was split 80%/20% for training and validation. Following Rusak et al. (2020), Gaussian noise was added to enhance generalization. For

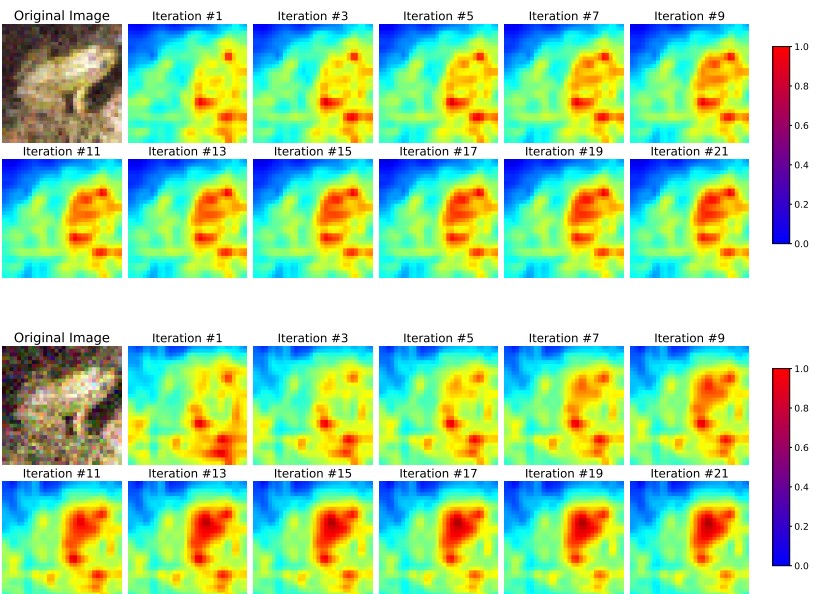

Figure 11: Visualizing the model's sequential reasoning process on CIFAR10-C with different corruption levels (level 1 and level 5).

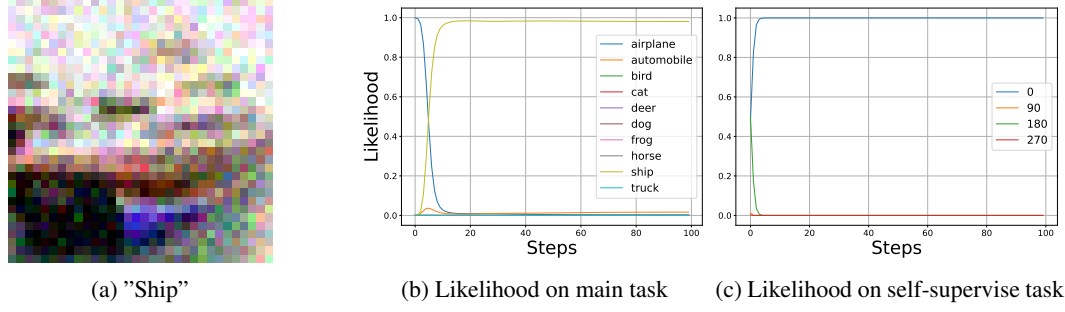

(a) "Ship"      (b) Likelihood on main task   (c) Likelihood on self-supervise task

Figure 12: Likelihood across iterations of a level 5 Gaussian Noise sample in class "Ship"

self-supervised learning, each input image was randomly rotated using a function $f_{rotate}$ selecting from $\{0°, 90°, 180°, 270°\}$, resulting in the augmented input $x = f_{rotate}(\text{clip}(x + \delta))$, with $\delta \sim \mathcal{N}(0, \sigma^2 \mathbf{I})$, $\sigma = 0.04$ (corresponding to level-1 Gaussian corruption in CIFAR-C), and the result is clipped to $[0, 1]^N$.

All experiments were conducted using an NVIDIA RTX 3090 GPU with 24GB VRAM.

## A.7 ANALYSIS OF SELF-SUPERVISED ITERATION ESTIMATION

To further characterize the relationship between the auxiliary self-supervised task and the main classification objective, we examine both per-sample likelihood trajectories (Figures 12a–13c) and batch-level correlation statistics (Table 4).

For images with strong directional cues, such as the "Ship" example, the auxiliary rotation predictions remain stable across iterations and closely follow the main-task likelihood trajectory. This alignment enables accurate identification of the optimal inference iteration. In contrast, for samples with weak edges or isotropic texture (e.g., "Cat"), the auxiliary predictions become unstable and may peak at an incorrect rotation, even as the main-task likelihood cleanly converges to the true label. These cases illustrate that the two tasks can decouple when orientation cues are inherently ambiguous.

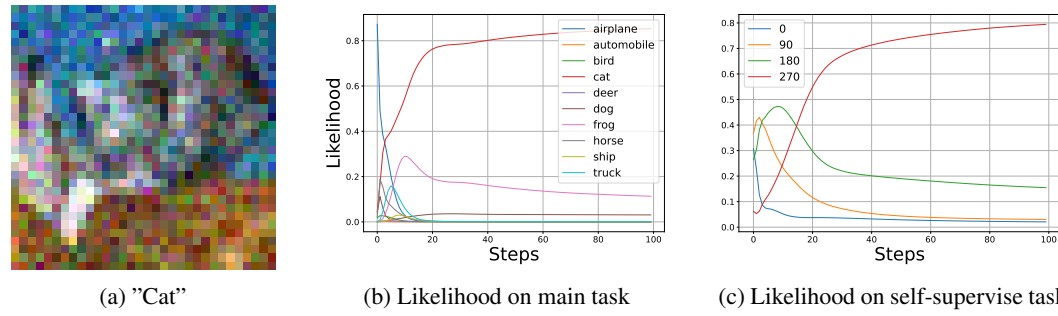

(a) "Cat"      (b) Likelihood on main task      (c) Likelihood on self-supervise task

Figure 13: Likelihood across iterations of a level 5 Gaussian Noise sample in class "Cat"

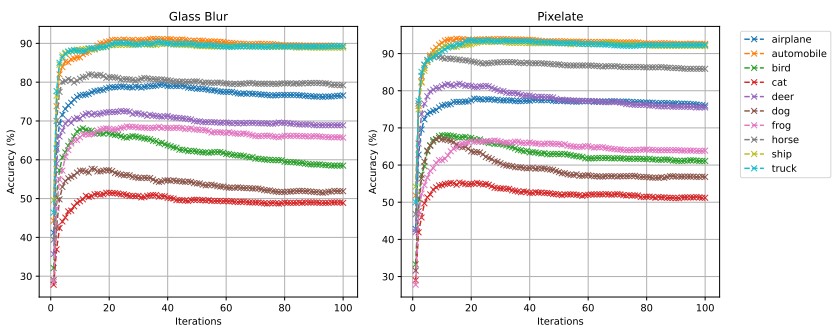

Figure 14: Self-supervised accuracy per class on CIFAR10-C, severity level 5.

Although per-sample decoupling appears in such ambiguous cases, batch-level behavior remains remarkably stable. Figure 14 shows that auxiliary accuracy curves are smooth and consistent across classes, even when some individual examples behave irregularly. This suggests that while the auxiliary task may mispredict specific samples, it still captures the global trend of the model's inference dynamics.

This observation is quantitatively confirmed in Table 4, which reports Pearson and Spearman correlations (mean ± std over three seeds) across 15 corruption types. Across the CIFAR10-C benchmark, 11 out of 15 corruptions show high Pearson/Spearman correlations between stepwise auxiliary and main accuracies, supporting Assumptions A.1 and A.2.

Four corruption types—Contrast, Defocus Blur, Motion Blur, and Zoom Blur—produce noticeably lower or noisy Spearman correlations. These are precisely the corruptions that reduce or destroy directional structure. Without meaningful orientation cues, rotation prediction cannot reliably track the main task, causing the auxiliary and main trajectories to diverge (as in the "Cat" example). We acknowledge this as an inherent limitation of using rotation prediction as a universal self-supervised proxy. Although, across all 15 CIFAR10-C corruptions, the average accuracy gap is only 0.984%, i.e., below 1%, demonstrating that the method is effective even under challenging perturbations. This suggests substantial room for future improvement with alternative auxiliary objectives or multimodal extensions.

## A.8 ACCURACY RESULTS ON OOD DATASETS

Tables 5 and 6 present the peak accuracy (%) of ViT, ResNet-30, Conv-LiGRU, and Conv-GRU on CIFAR10-C and CIFAR100-C datasets at corruption level 5, respectively. The results are based on the first random seed.

Across nearly all corruption types, Conv-GRU consistently outperforms the other models, demonstrating superior robustness to various corruptions. Conv-LiGRU generally performs better than ResNet-30 and ViT, highlighting the advantage of recurrent deep thinking architectures.

Table 4: Full correlation analysis across all 15 corruption types (CIFAR10-C, severity 5). For each corruption, we report Pearson and Spearman correlations (mean $\pm$ std over 3 seeds) between the stepwise auxiliary accuracy and the main-task accuracy, together with the mean accuracy gap between the auxiliary-selected peak and the true peak.

| Corruption | Pearson | Spearman | Acc Gap (%) |
|---|---|---|---|
| Gaussian Noise | $0.92 \pm 0.04$ | $0.68 \pm 0.20$ | 1.64 |
| Shot Noise | $0.90 \pm 0.03$ | $0.76 \pm 0.08$ | 1.12 |
| Impulse Noise | $0.88 \pm 0.08$ | $0.73 \pm 0.13$ | 2.18 |
| Defocus Blur | $0.89 \pm 0.14$ | $0.48 \pm 0.39$ | 0.61 |
| Glass Blur | $0.93 \pm 0.03$ | $0.95 \pm 0.03$ | 0.14 |
| Motion Blur | $0.91 \pm 0.08$ | $0.46 \pm 0.23$ | 1.36 |
| Zoom Blur | $0.90 \pm 0.09$ | $0.50 \pm 0.48$ | 1.11 |
| Snow | $0.93 \pm 0.02$ | $0.86 \pm 0.04$ | 0.52 |
| Frost | $0.95 \pm 0.02$ | $0.94 \pm 0.06$ | 0.30 |
| Fog | $0.93 \pm 0.02$ | $0.49 \pm 0.38$ | 1.18 |
| Brightness | $0.93 \pm 0.04$ | $0.77 \pm 0.17$ | 0.53 |
| Contrast | $0.55 \pm 0.32$ | $-0.07 \pm 0.27$ | 2.99 |
| Elastic Transform | $0.95 \pm 0.02$ | $0.83 \pm 0.17$ | 0.53 |
| Pixelate | $0.95 \pm 0.01$ | $0.88 \pm 0.07$ | 0.49 |
| JPEG Compression | $0.97 \pm 0.01$ | $0.91 \pm 0.11$ | 0.06 |

Table 5: Peak accuracy (%) of ViT, Resnet-30, Conv-LiGRU and Conv-GRU on CIFAR-C, level 5.

| | Gauss | Shot | Impulse | Defocus | Glass | Motion | Zoom | Snow | Frost | Fog | Bright | Contrast | Elastic | Pixelate | JPEG | Avg. |
|---|---|---|---|---|---|---|---|---|---|---|---|---|---|---|---|---|
| ViT | 57.0 | 57.8 | 55.1 | 59.2 | 56.9 | 56.8 | 56.5 | 50.8 | 50.6 | 27.5 | 52.5 | 21.9 | 57.0 | 59.1 | 57.8 | 51.8 |
| Resnet | 72.9 | 73.5 | 55.6 | 53.3 | 60.3 | 51.8 | 58.7 | 73.4 | 72.3 | 43.3 | 79.2 | 19.9 | 62.2 | 62.2 | 71.6 | 60.7 |
| CLiGRU | 64.2 | 73.2 | 54.5 | 57.9 | 61.7 | 58.1 | 63.2 | 73.7 | 73.9 | 42.8 | 79.7 | 18.0 | 68.4 | 64.4 | 77.1 | 62.1 |
| CGRU | **77.2** | **77.1** | **59.0** | **62.0** | **62.4** | **63.8** | **67.7** | **77.7** | **77.4** | **54.6** | **82.4** | **28.7** | **70.8** | **66.4** | **78.9** | **67.1** |

Notably, the improvements of Conv-GRU are more pronounced on CIFAR10-C than on CIFAR100-C, which is expected due to the increased complexity and number of classes in CIFAR100. These findings reinforce the efficacy of deep thinking models in handling distribution shifts with better generalization compared to standard feed-forward and transformer models.

### A.9 OVERTHINKING AND ESTIMATED RESULTS BY SELF-SUPERVISED TASK

15 illustrates the correlation between accuracy curves of the main task and the auxiliary self-supervised task (Conv-GRU) across 15 corruption types in OOD datasets CIFAR10-C (a) and CIFAR100-C (b). The dark blue shaded region marks the estimated recurrent steps before overthinking, as determined by our proposed algorithm.

We observe that the auxiliary task accuracy closely tracks the main task accuracy during the early iterations, enabling reliable estimation of the optimal stopping point without access to labels. This strong correlation holds consistently across different corruptions and datasets, confirming the auxiliary task as an effective proxy for peak iterations detection in deep thinking models under distributional shifts.

The visualizations also highlight the onset of overthinking, where both accuracies begin to degrade beyond the shaded region, demonstrating the practical value of our method in dynamically estimating suitable iterations to avoid performance drops on OOD data.

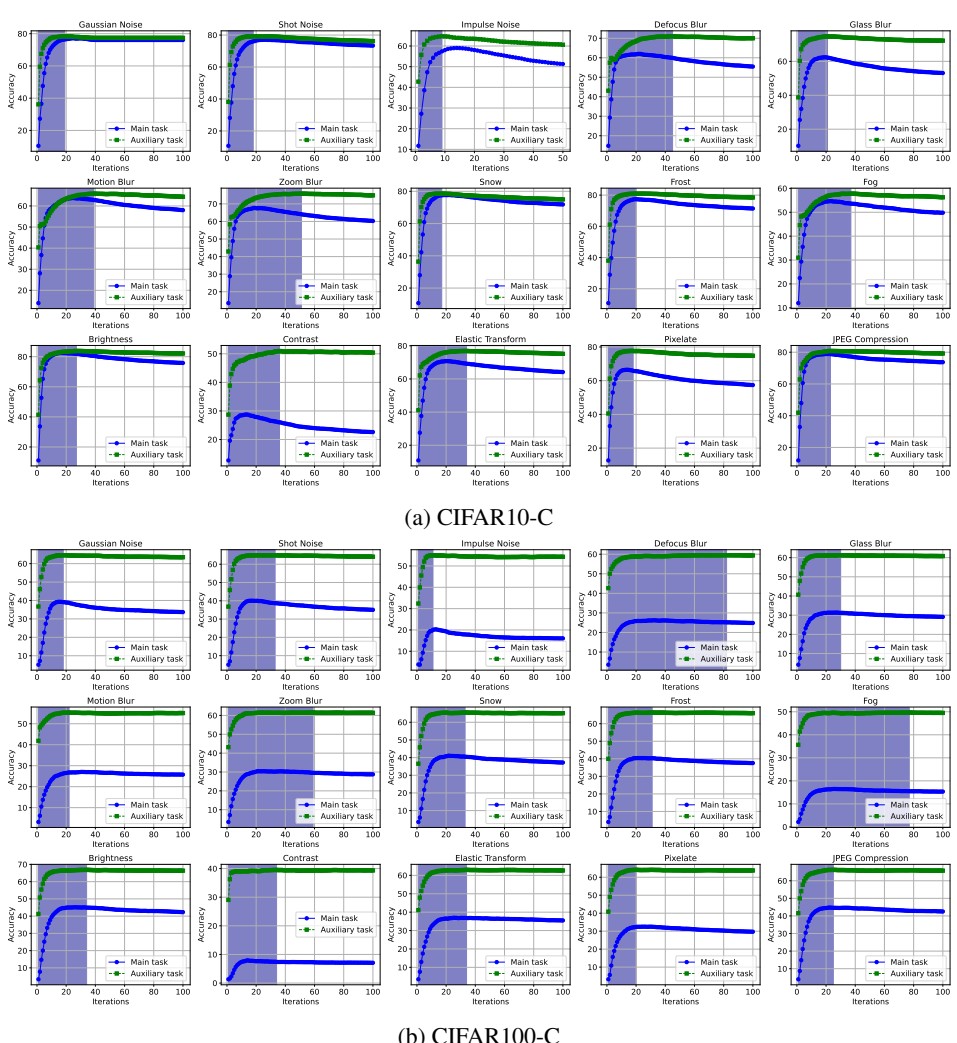

Figure 15: The accuracy correlation of the main and auxiliary tasks of Conv-GRU on OOD datasets. The dark blue area is the recurrent steps before overthinking, estimated by our algorithm 1.

Table 6: Peak accuracy (%) of ViT, Resnet-30, Conv-LiGRU and Conv-GRU on CIFAR100-C, level 5.

| | Gauss | Shot | Impulse | Defocus | Glass | Motion | Zoom | Snow | Frost | Fog | Bright | Contrast | Elastic | Pixelate | JPEG | Avg. |
|---|---|---|---|---|---|---|---|---|---|---|---|---|---|---|---|---|
| ViT | 31.2 | 31.2 | **28.6** | **32.7** | 31.0 | **30.8** | **31.3** | 25.7 | 25.4 | 8.2 | 25.1 | 5.0 | 32.1 | **33.0** | 31.6 | 26.8 |
| Resnet | 27.0 | 28.1 | 13.5 | 21.8 | 21.3 | 19.5 | 23.7 | 26.5 | 26.5 | 11.6 | 31.3 | 4.6 | 24.6 | 27.1 | 31.6 | 22.6 |
| CLiGRU | 32.4 | 34.9 | 16.4 | 26.0 | 28.6 | 24.9 | 29.1 | 33.3 | 33.4 | 12.6 | 39.8 | 4.7 | 32.5 | 29.9 | 39.2 | 27.9 |
| CGRU | **39.2** | **40.0** | 20.3 | 26.0 | **31.4** | 27.0 | 30.4 | **41.1** | **40.3** | **16.4** | **45.1** | **7.9** | **37.0** | 32.4 | **44.8** | **31.9** |

