# OpenReview forum: "Deep Thinking on Out-Of-Distribution Data: How can we know when a model is overthinking?"
_ICLR.cc/2026/Conference — Submitted to ICLR 2026_

### Official Review · Reviewer_pdNM · 2025-10-26

**Soundness:** 2
**Presentation:** 3
**Contribution:** 2
**Rating:** 4
**Confidence:** 3

**Summary:**

This paper analyses how existing recurrent models with flexible compute can outperform higher parameterized models (aka resnet, ) in ood settings like classification on datasets like cifar etc, and its corrupted variants.

**Strengths:**

- The paper explores the extrapolation capabilities of deep thinking architectures, particularly when they are allowed to think in the latent space for a long number of iterations.
- Authors show that ACT leads to underthinking which can be solved by their proposed SSL approach.

**Weaknesses:**

I have several concerns about the paper as noted below:
- concerns on novelty: it seems that the paper combines two well known ideas 1) the fact that recurrent-thinking models can lead to strong OOD performance, if dedicated more compute during inference (original paper titled Scaling up Test-Time Compute with Latent Reasoning: A Recurrent Depth Approach). 2) test-time-training. for eg, the efficiency of rotation prediction task as a surrogate ssl task is already well known(TTT-MAE).
- the application of ttt + recurrent thinking seems interesting, however, i believe that that it still raises some novelty concerns for me.
- conv-gru/conv-ligru: fig 2 led me to believe that convli-gru is a novel architecture , and should perform better than conv-gru. however, it appears that convli-gru underperforms in table 1. so i am not clear on why convli-gru is proposed as a method in the first place.
- needs more results on other datasets: the paper reports results on cifar10. however, following hendryks et al, and existing literature in test-time-training, results on bigger datasets like imagenet-v2, imagenetsketch, imagenet-a, imagenet, and 10 fine-grained datasets should be reported.
- algorithm 2 (Step 4) seems largely similar to Avi's paper(original paper titled Scaling up Test-Time Compute with Latent Reasoning: A Recurrent Depth Approach). , and is already a well known step in the theory of recurrent models.
-  assumption A1, A2 (lines 208-211) appears a big limitation of method: how can one know that for a given task, dataset, and model, which ssl task will hold a direct correlation with the downstream accuracy. The premise of the paper lies on this assumption, so it would be beneficial to provide some analysis on when this assumption might hold.
- what happens if the network is ran till t= infinity ?

**Questions:**

please see weaknesses.

---

> ### Author Response · Authors · 2025-11-15
> **Answer for Weakness 1, 2, 5**
>
> # Weakness 1, 2, 5:
> - "concerns on novelty: it seems that the paper combines two well known ideas ..."
> - "the application of ttt + recurrent thinking seems interesting, ..."
> - "algorithm 2 (Step 4) seems largely similar to Avi's paper..."
>
> # Anwser
>
> Thank you for raising the question regarding novelty. First, we would like to clarify our work here:
>
> Initially, we aimed to explore the extrapolation capabilities of deep thinking (recurrent-depth) architectures for object recognition. In doing so, we observed that increasing the number of recurrent iterations at test time can improve OOD accuracy, and that harder, more corrupted inputs naturally require more iterative refinement—indicating a reasonable inference behavior.
>
> However, we also discovered an issue: overthinking, which was reported by Bansal et al. (2022) in logical tasks. After a certain point, accuracy begins to degrade when too many iterations are executed. This makes test-time computation beneficial only up to an unknown, input-dependent limit.
>
> To address this, we propose using a lightweight self-supervised proxy task to detect when performance starts to decline and to estimate the optimal reasoning depth without any parameter updates. Unlike halting mechanisms such as ACT, this approach does not restrict the reasoning capacity of recurrent models, thereby avoiding underthinking, while providing a data-driven way to determine when further iterations become harmful.
>
>
> We now would like to clarify **the conceptual distinctions between our work and prior research**.
>
> ### 1. Our setting differs fundamentally from prior “recurrent test-time compute” work
>
> The paper mentioned by you (Scaling up Test-Time Compute with Latent Reasoning) addresses autoregressive NLP generation, where additional computation is injected in the latent space before predicting the next token.
>
> In contrast:
>
> Our architecture is inspired by Deep Thinking (Bansal et al., 2022) and Neural GPU (Kaiser & Sutskever, 2015), where the entire input is repeatedly processed by a recurrent block. This is fundamentally different from the mechanism in “Scaling up Test-Time Compute with Latent Reasoning”, where only the hidden state is iteratively refined at each decoding step in an autoregressive NLP setting. Moreover, that line of work does not study overthinking, visual corruption, or test-time scaling in vision.
> Thus, despite superficial similarity in “test-time compute,” the task, mechanism, and problem formulation are fundamentally different.
>
> ### 2. Our method does not use TTT, nor TTT-MAE–style adaptation
>
> Although we employ a self-supervised task, we do not update model parameters at test time. The auxiliary rotation task is used solely as a proxy to estimate when overthinking begins, unlike TTT-MAE or other TTT methods that adapt the representation. **Our approach is a test-time iteration selection mechanism, not test-time training**.
>
> ### 3. Novel empirical finding: overthinking in recurrent-thinking models for vision
>
> To the best of our knowledge, our paper is the first to show that deep recurrent-thinking models suffer from overthinking in computer vision: We demonstrate this phenomenon consistently across 15 corruptions and multiple datasets. We also show that hidden-state convergence does not prevent post-peak accuracy decay, a behavior not documented in previous work. This provides a new lens for understanding iterative reasoning in vision models. This is a new problem and motivates the need for a new solution.
>
> ### 3. Clarification on Algorithm 2
>
> We fully agree with you that Algorithm 2 resembles steps common in recurrent model training.
>
> Algorithm 2 is the Progressive Loss method introduced by Bansal et al. (2022), which we reproduce only for completeness in Appendix A.4. This algorithm aims to enhance the qualification of all steps in recurrent thinking models. From that, make models avoid degrading accuracy when expanding the number of reason iterations.
>
> **Our contribution does not rely on Progressive Loss**.
>
> In fact, we explicitly show in Figure 1 that even when Progressive Loss is applied, overthinking still occurs, highlighting the need for our new self-supervised estimator.
>
> Thus, Algorithm 2 is included to contextualize related approaches, not as a component we contribute or claim as novel.
>
> ### 4. Our novelty:
>
>  - We explore the extrapolation capabilities of deep thinking architectures for object recognition. We show that extending the Thinking Process enables strong generalization in OOD settings as a form of test-time scaling.
>
>  - We introduce a novel self-supervised approach to detect the onset of *overthinking* and adaptively select the optimal reasoning depth, moving beyond strict halting rules that often induce
>
> We hope this clarifies that our contribution goes beyond combining existing techniques, and instead introduces a new perspective on test-time inference behavior of recurrent-thinking models in computer vision.

---

> ### Author Response · Authors · 2025-11-17
>
> # Weakness 6:
>
> assumption A1, A2 (lines 208-211) appears a big limitation of method: how can one know that for a given task, dataset, and model, which ssl task will hold a direct correlation with the downstream accuracy. The premise of the paper lies on this assumption, so it would be beneficial to provide some analysis on when this assumption might hold.
>
> # Answer:
>
> Thank you for raising this important point about the conditions under which Assumptions A.1 and A.2 hold. Since related concerns were also expressed by the other reviewers, we have provided a detailed quantitative analysis, visualizations in the revised Appendix 7, as well as in our response to **Reviewer XTzY**, where we include the full Pearson/Spearman correlation table. Here, we summarize the key observations.
>
> Our analysis shows that these assumptions tend to hold when the auxiliary rotation-prediction task preserves structural cues that are also informative for the main classification task. As illustrated in Appendix 7, samples with clear directional features (e.g., the “Ship’’ example, figure 12, appendix 7) exhibit auxiliary trajectories that evolve in close alignment with the main-task likelihood, whereas samples with weak or isotropic textures (e.g., the “Cat’’ example, figure 13, appendix 7) may display unstable or mismatched auxiliary peaks, reflecting potential per-sample decoupling under ambiguous orientations.
>
> Crucially, even when such per-sample mismatches occur, the batch-level behavior remains stable: the auxiliary accuracy curves are smooth, consistent across classes, and continue to track the global inference dynamics of the main task (Figure 14, appendix 7). This observation is reinforced by our full correlation study (Table 4), where 11 out of 15 CIFAR10-C corruptions exhibit strong Pearson/Spearman correlations. The remaining cases correspond precisely to corruptions that remove directional structure. Despite these challenging conditions, the average accuracy gap across all corruptions remains below 1%, indicating that the auxiliary task still reliably captures the iteration-wise trend needed for peak selection.
>
> We hope this analysis clarifies the regimes in which A.1 and A.2 are most likely to hold. We kindly refer the reviewer to Appendix 7 and our detailed response to Reviewer XTzY for the complete evidence.

---

> ### Author Response · Authors · 2025-11-18
>
> # Weakness 3:
> - conv-gru/conv-ligru: fig 2 led me to believe that convli-gru is a novel architecture , and should perform better than conv-gru. however, it appears that convli-gru underperforms in table 1. so i am not clear on why convli-gru is proposed as a method in the first place.
>
> # Answer:
>
> Thank you for raising this point. We apologize for the confusion. Our intention was not to present Conv-LiGRU as a stronger architecture than Conv-GRU, nor as a novel contribution in itself. Conv-LiGRU is simply a convolutional adaptation of the LiGRU block from Ravanelli et al. (2018), replacing the fully connected transformations with convolutional ones so that it fits the vision-oriented recurrent “deep thinking’’ framework.
>
> Our goal in including Conv-LiGRU was to test whether overthinking behavior and our proposed estimator generalize across different recurrent block designs, rather than being specific to Conv-GRU. In fact, as discussed in Section 5, Conv-LiGRU tends to exhibit weaker overthinking than Conv-GRU, which provides an informative contrast across architectures.
>
> Regarding Table 1, the results reported are the peak accuracies achieved when no limit is placed on the number of recurrent iterations. Under this unrestricted setting, Conv-GRU attains a slightly higher maximum accuracy, even though it overthinks more severely. This explains why Conv-GRU appears stronger in Table 1 despite the qualitative differences shown in Figure 3.
>
> We appreciate the reviewer pointing out this potential source of confusion and will clarify the motivation and interpretation of Conv-LiGRU in the revised version.

---

> ### Comment · Reviewer_pdNM · 2025-11-19
> **reply**
>
> dear authors,
> thanks for the rebuttal, and valuable comments. i apologize for delay in my reply, got lost in my own cvpr submission, and iclr rebuttals lol :-).  my own reviewers are brutal this time :-(.
>
>  coming back to the topic, i will be grateful for some clarifications:
>
> `we do not update model parameters at test time. `
> just confirming, line 209/210: require training auxiliary head during training right? and you guys freeze it during testing, and keep track of running accuracy? that still requires training an aux head,
>
> if the network is not trained with aux head, how will it work? that seems to be a limitation.?
>
> `on fundamental intuition behind overthinking`
>
> i saw your pearson plots bw aux task acc vs actual acc. they look pretty cool. my key question is why overthinking is a problem? intuitively, what happens if net is ran till `t = infinity`. is the representation absolutely destroyed?
>
> is there a case possible, when aux task acc is pretty high, but downstream task is lower accuracy.
>
> `on fixed points in deep equilibrium models`
>
> there has been work on deep equilibrium models by zico kotler (and team) which focuses on predicting `stable representations` even if the residual block is ran till infinity,
>
> if one accepts that premise, then the overthinking is not a problem. (since final representation reaches a stable minima, and fixed convergence point, more iterations wont hurt). whats your take on that?
>
> now i agree, that overthinking might be an issue in gru/lstm/iterative self attention type nets, but not in deqs.
>
> `our paper is the first to show that deep recurrent-thinking models suffer from overthinking in computer vision`
> https://openreview.net/pdf?id=nwDRD4AMoN
>
> fig 2 of this paper follows a similar recurrent architecture, fig 8 provides results on computer vision (corrupted test samples), so they also talked about overthinking (aka increasing the test-time-compute iterations), they also report degradation in results (fig 6c)
>
> i agree their fig 2 does not use the input during `every` recurrent iteration, in the same sense as  `deep thinking nets`, but i think that is a design choice, and not a fundamental architecture difference (unless of course you wanna retain memory).
>
> `needs more results.`
> the paper focuses on ood classification tasks (line 87-89). i believe, a comprehensive testing including imagenet, imagenet v2, imagenet-sketch, 10 fine grained datasets (like pets etc.) is required before making conclusive claims.
>
> i understand paper's scope is not ttt/tda, but i believe these results will greatly boost the paper quality.
>
> `line 209-210`
> why is input rotated by only 4 angles. why not continuous? is angle prediction a regression problem or classification problem (1/4 classes)? what happens if we do regression.
>
> best,
>
> reviewer

---

> ### Author Response · Authors · 2025-11-21
>
> Thank your insightful comment, we will continue to reply to your questions above here:
>
> ---
> - **Question**: *if the network is not trained with aux head, how will it work? that seems to be a limitation.?*
> - **Answer**: The key idea of our proposal method is that using the aux task (a self-supervised task) to monitor the main task accuracy, from that detect the onset of overthinking, and approximate the peak accuracy iteration. So, our method requires the model to be trained with an auxiliary task. We think it is a dual advantage, and not a limitation. Besides the benefit for our algorithm, as we mentioned in Section 3.2, following the work by Balaji et al. (2018) on *Metareg: Towards domain generalization using meta-regularization*, and Zeyu Feng et al 2019 on *Self-Supervised Representation Learning by Rotation Feature Decoupling*, we treat the auxiliary task as a regularization task, which also encourages the model to learn invariant features that help it generalize across distribution shifts.
>
> ---
>
> - **Question**:
>   - `on fundamental intuition behind overthinking`
> i saw your pearson plots bw aux task acc vs actual acc. they look pretty cool. my key question is why overthinking is a problem? intuitively, what happens if net is ran till t = infinity. is the representation absolutely destroyed?
>   - `on fixed points in deep equilibrium models`
> there has been work on deep equilibrium models by zico kotler (and team) which focuses on predicting stable representations even if the residual block is ran till infinity,
> if one accepts that premise, then the overthinking is not a problem. (since final representation reaches a stable minima, and fixed convergence point, more iterations wont hurt). whats your take on that?
> now i agree, that overthinking might be an issue in gru/lstm/iterative self attention type nets, but not in deqs.
>
> - **Answer**:  \
>   Your questions are absolutely interesting.
>   - Many prior studies assumed that the representation would converge when $t == infinity$, and some works (as `MIND over body: Adaptive thinking using dynamic computation`  (Mathur et al., 2025), we mentioned in related work (section 2.2), based on that, to halt recurrent iteration when the difference between successive hidden states is below a threshold. But, we asked a question about how we deal with overthinking, even when the representation is not convergence? So, this is our self-supervised estimation algorithm that tries to solve; we did not add any constraints to force models to fix-point convergence, and so, we try to solve overthinking and underthinking most naturally by a self-supervised estimator.
>   - We also have another concern when working on this work is that:
> > **Is the representation at $\( t \to \infty \)$ actually the best representation produced by a recursive model?**
>
>     As we discuss in *Section 5 (Contraction Assumption and Overthinking)*, our experiments show that even when the hidden state $\( h_t \)$ appears to converge, Conv-GRU still overthinks. Model accuracy rises, reaches a peak at some intermediate iteration, and then degrades as inference continues. This behavior demonstrates that the limiting representation $\( h_{t \to \infty} \)$ is *not guaranteed to be the optimal representation*. Instead, the best representation often occurs at a *finite iteration*, after which additional recursive computation becomes harmful. We also know that DEQs defined infinite depth models based on the assumption of fixed-point iteration, but as we discussed above, infinite representation can not guarantee the model achieves peak accuracy. So, from our perspective, deep thinking models should be trained naturally, without any constraints; this is the way to explore the full potential of these kinds of models.
>
> ---

---

> ### Author Response · Authors · 2025-11-21
>
> - **Question**:
>   - ``our paper is the first to show that deep recurrent-thinking models suffer from overthinking in computer vision`` https://openreview.net/pdf?id=nwDRD4AMoN
>   - fig 2 of this paper follows a similar recurrent architecture, fig 8 provides results on computer vision (corrupted test samples), so they also talked about overthinking (aka increasing the test-time-compute iterations), they also report degradation in results (fig 6c)
>   -i agree their fig 2 does not use the input during every recurrent iteration, in the same sense as deep thinking nets, but i think that is a design choice, and not a fundamental architecture difference (unless of course you wanna retain memory).
>
> - **Answer**:
>   - Thank you for pointing out the observations in Miyato et al. (2025). We agree that their Fig. 6c and Figs. 16–17 show cases where performance decreases when the number of recurrent steps increases. However, these occurrences are incidental to their main contributions and are not analyzed as a dedicated phenomenon. The paper does not examine hidden-state dynamics, does not connect convergence to degradation, and does not study overthinking systematically across computer-vision benchmarks.
>
>     In contrast, our paper focuses specifically on characterizing overthinking as a phenomenon, analyzing how accuracy peaks and then declines even when $h_t$ converges, and demonstrating this behavior consistently across multiple corruption types and a real distribution shift (CIFAR→STL10). The prior work therefore documents isolated instances of degradation, but does not provide a systematic study of overthinking as we do here.
> ---
> - **Question**:
>   - why is input rotated by only 4 angles. why not continuous? is angle prediction a regression problem or classification problem (1/4 classes)? what happens if we do regression.
> - **Answer**:
>   - Thank you for the insightful suggestion. We agree that using a continuous rotation and formulating the auxiliary task as a regression problem is an interesting idea, especially given that our analyses (see Appendix A.7) show that the per-sample auxiliary likelihood can be unstable for images with weak or isotropic structure. A continuous-angle regression objective might indeed help stabilize the signal at the sample level and potentially improve per-sample iteration estimation.
>
>     In this work, we follow prior rotation-prediction literature (Sun et al., 2020; Zeyu Feng et al., 2019) and use the canonical four discrete angles (0°, 90°, 180°, 270°). Besides aligning with prior practice, these four rotations preserve the input size exactly without requiring padding or interpolation, which keeps the implementation simple and ensures consistent input geometry across iterations.
>
>     We appreciate the reviewer’s suggestion, and exploring continuous-angle regression as a stronger auxiliary signal is a promising direction for future work.

---

> ### Comment · Reviewer_pdNM · 2025-11-23
>
> i thank the authors for their detailed reply,
> I still believe that the paper `needs more results (as mentioned in the comment above)`, and more empirical verification before some of the claims can be made,
>
> i will wait till the discussion window , as mentioned by the authors in the global comment,
>
>
> sincerely,
> reviewer

---

> > ### Author Response · Authors · 2025-11-23
> >
> > Oh, sorry. We missed replying this comment **needs more results (as mentioned in the comment above)**. We are trying and activing more experiments on coco2017 dataset (we described in the global comment above), we will supply new results immediately after completing these experiments.

---

### Official Review · Reviewer_XTzY · 2025-10-29

**Soundness:** 2
**Presentation:** 3
**Contribution:** 2
**Rating:** 2
**Confidence:** 4

**Summary:**

The authors propose a formalism for studying _overthinking_ in recurrent _deep-thinking_ vision models under covariate, showing that test accuracy peaks and then decreases as the number of _thinking_ iterations increases. This aligns with prior observations on algorithmic tasks, but is shown repeatedly through multiple experiments on CIFAR-C and Tiny-ImageNet-C (eg: Fig 1, Fig 6). To assess overthinking, the authors introduce a label-free test-time selection rule (Algorithm 1) via the self-supervised auxiliary task of rotation prediction. They then demonstrate the usefulness of the proposed rule by direct methodological contrast against ACT and norm thresholds (Fig 7, Table 3) and further show that iterative test-time scaling can help OOD robustness with fewer parameters as compared to feed-forward models (Tables 1, 2, 4, 5).

**Strengths:**

- The key idea behind the selection rule of the paper is to pick the iteration step to stop at that corresponds to peak accuracy on the auxiliary task, with the main contribution being that it's label-free unlike existing inference-time adaptation methods. This is simple, and potentially domain-agnostic for inference-time adaptation without labels, unlike existing methods such as ACT and norm-thresholds, which the proposed method is either competitive or stronger than while being label-free.
- From my limited knowledge, only _Bansal et al. (2022)_ actually demonstrated accuracy peaking and then declining with further recurrent steps, and exclusively on synthetic algorithmic reasoning tasks (e.g., addition, sorting). None of the vision-related works they cite (e.g., (Graves, 2016; Veerabadran et al., 2023; Mathur et al., 2025) a) report such degradation. If true, this position can be strengthened by explicitly clarifying that the previous algorithmic tasks denote symbolic sequence computations and not ML datasets.

**Weaknesses:**

- The central assumption (stated in A.1 and A.2) is untested: the curves in Figs 6 and 11 are only qualitative. Plus, the naming is misleading: they are NOT showing correlations, they are only reporting accuracies. Nowhere in the paper could I find the correlation coefficients, associated confidence intervals, variation under or outlier (failure) cases where rotation might be ambiguous. Concrete way to address this issue: report Pearson/Spearman correlation coefficients between stepwise auxiliary and main accuracies for each corruption (mean ± s.d. over seeds) with any comments on where it fails.
- I am unclear on the details on training parity between the models: were BN, RandAugment, EMA, label smoothing etc used for training the feed-forward CNNs from scratch, was ViT trained with augmentations? These are important since these architectural changes are known to achieve higher CIFAR-C accuracy and show better OOD robustness. So, this would make the claim of Conv-GRU/Conv-LiGRU convincing.
- It is also not clear if the recurrent models are benefitting from extra compute at inference time, can you show a comparison that is FLOPs-matched?
- In Lemma 1, how do we know that the task loss shrinks multiplicatively? Doesn't this need to be derived from the dynamics? We need the state update map to shrink thus and then be able to make a statement about the loss if F is derived from the gradient of the loss. The lemma seems to be confusing hidden-state dynamics with the functional used to measure performance and hence skipping an important step. Also some explanation of the "bounded by non-monotonic" loss function in this context would be helpful. It is possible I am missing something and I am happy to reconsider in that case else linking the dynamics and the loss functional would make the lemma meaningful.
- When looking at the curve comparing accuracy and norm difference (such as Fig 3), the connection with overthinking is less conspicuous. The ranges of the x axes across these plots (for eg vs. Fig 6) are vastly different (40 vs 100 where peak is at 25). Is it possible to make the comparison clearer such as by having consistent regions of overthinking?
- The y axes for the two plots shown side by side in Figure 7 have very different scale: each unit means 10 for the plot on the left, and 5 for the plot on the right. For the argument about the unhalted model surpassing ACT on STL-10 to be more convincing, having similar scales will help.


I am happy to reconsider my score if my issues and questions can be addressed.

**Questions:**

- Re: the auxiliary task of rotation prediction, from my understanding, the paper implicitly assumes that both the main and the auxiliary tasks depend on similar representation quality whereas for some classes, the rotation prediction task might be intrinsically ambiguous (such as a bottle, ball, donut, frog, ship etc). For these, rotation prediction accuracy curve may not peak at the same iteration as the main classification curve-- it might saturate early or fluctuate after a certain number of iterations. Do you observe this, and how relevant is it towards the central assumption of the paper? Are there positive correlation coefficients between the two tasks for classes with rotational symmetry as well? Both CIFAR-C and Tiny-ImageNet-C consist of such classes. Concretely, can you report per-class correlation showing trends between asymmetric and symmetric classes?
- To test the scope of the OOD claim beyond narrow, _artificially corrupted_ benchmarks such as CIFAR-C and Tiny-ImageNet-C that capture low-level pixel perturbations but not true domain or semantic shifts, what do you think about testing on benchmarks such WILDS, PACS etc? This would reveal whether the rotation-proxy criterion and iterative test-time scaling extend to _real distribution shifts_ rather than only to synthetic corruptions.

---

> ### Author Response · Authors · 2025-11-16
>
> # Comment:
>
> From my limited knowledge, only Bansal et al. (2022) actually demonstrated accuracy peaking and then declining with further recurrent steps, and exclusively on synthetic algorithmic reasoning tasks (e.g., addition, sorting). None of the vision-related works they cite (e.g., (Graves, 2016; Veerabadran et al., 2023; Mathur et al., 2025) a) report such degradation. If true, this position can be strengthened by explicitly clarifying that the previous algorithmic tasks denote symbolic sequence computations and not ML datasets.
>
> # Answer:
>
> Thank you for the helpful observation. We agree with you that prior reports of accuracy peaking and post-peak degradation have primarily arisen in synthetic or algorithmic reasoning settings (e.g., addition, sorting, maze/pathfinding). To the best of our knowledge, no previous work has demonstrated overthinking on standard computer-vision datasets or corruption benchmarks.
>
> Within our understanding of the literature, our work is therefore the first to systematically document overthinking dynamics on natural-image tasks (e.g., CIFAR10-C, TinyImageNet-C, STL-10) and to show that recurrent reasoning can degrade accuracy across a wide range of realistic visual corruptions.
>
> If helpful, we are happy to clarify this distinction explicitly in the revised version of the paper.

---

> ### Author Response · Authors · 2025-11-16
>
> Cause the limitation of words in each comment, we first show the table of Pearson and Spearman correlations between the stepwise auxiliary accuracy and the main-task accuracy.
>
> Full correlation analysis across all 15 corruption types (CIFAR10-C, severity 5).
> For each corruption, we report Pearson and Spearman correlations (mean $\pm$ std over 3 seeds)
> between the stepwise auxiliary accuracy and the main-task accuracy, together with the mean accuracy gap
> between the auxiliary-selected peak and the true peak.
>
> | **Corruption**        | **Pearson**        | **Spearman**        | **Acc Gap (mean %)** |
> |-----------------------|--------------------|----------------------|------------------|
> | Gaussian Noise        | 0.92 ± 0.04        | 0.68 ± 0.20          | 1.64             |
> | Shot Noise            | 0.90 ± 0.03        | 0.76 ± 0.08          | 1.12             |
> | Impulse Noise         | 0.88 ± 0.08        | 0.73 ± 0.13          | 2.18             |
> | Defocus Blur          | 0.89 ± 0.14        | 0.48 ± 0.39          | 0.61             |
> | Glass Blur            | 0.93 ± 0.03        | 0.95 ± 0.03          | 0.14             |
> | Motion Blur           | 0.91 ± 0.08        | 0.46 ± 0.23          | 1.36             |
> | Zoom Blur             | 0.90 ± 0.09        | 0.50 ± 0.48          | 1.11             |
> | Snow                  | 0.93 ± 0.02        | 0.86 ± 0.04          | 0.52             |
> | Frost                 | 0.95 ± 0.02        | 0.94 ± 0.06          | 0.30             |
> | Fog                   | 0.93 ± 0.02        | 0.49 ± 0.38          | 1.18             |
> | Brightness            | 0.93 ± 0.04        | 0.77 ± 0.17          | 0.53             |
> | Contrast              | 0.55 ± 0.32        | -0.07 ± 0.27         | 2.99             |
> | Elastic Transform     | 0.95 ± 0.02        | 0.83 ± 0.17          | 0.53             |
> | Pixelate              | 0.95 ± 0.01        | 0.88 ± 0.07          | 0.49             |
> | JPEG Compression      | 0.97 ± 0.01        | 0.91 ± 0.11          | 0.06             |

---

> ### Author Response · Authors · 2025-11-16
>
> # Weakness 1:
>
> The central assumption (stated in A.1 and A.2) is untested: the curves in Figs 6 and 11 are only qualitative. Plus, the naming is misleading: they are NOT showing correlations, they are only reporting accuracies. Nowhere in the paper could I find the correlation coefficients, associated confidence intervals, variation under, or outlier (failure) cases where rotation might be ambiguous. Concrete way to address this issue: report Pearson/Spearman correlation coefficients between stepwise auxiliary and main accuracies for each corruption (mean ± s.d. over seeds) with any comments on where it fails.
>
> # Question 1:
>
> Re: the auxiliary task of rotation prediction, from my understanding, the paper implicitly assumes that both the main and the auxiliary tasks depend on similar representation quality whereas for some classes, the rotation prediction task might be intrinsically ambiguous (such as a bottle, ball, donut, frog, ship etc). For these, rotation prediction accuracy curve may not peak at the same iteration as the main classification curve-- it might saturate early or fluctuate after a certain number of iterations. Do you observe this, and how relevant is it towards the central assumption of the paper? Are there positive correlation coefficients between the two tasks for classes with rotational symmetry as well? Both CIFAR-C and Tiny-ImageNet-C consist of such classes. Concretely, can you report per-class correlation showing trends between asymmetric and symmetric classes?
>
> # Answer:
>
> Thank you for the thoughtful and constructive comments. They significantly helped us improve both the clarity and rigor of the paper. In the revised version, we have added a dedicated discussion in Appendix 7, where we analyze in detail the correlation structure between the main task and the auxiliary self-supervised task, including per-sample visualizations and per-class trends as suggested.
>
> As illustrated by the two representative samples added in the revision (see Appendix 7, Figures 12 and 13), images with strong directional cues—such as the “Ship’’ example (Figure 12)—exhibit highly stable auxiliary rotation predictions that closely follow the main-task likelihood trajectory. In contrast, images with weak edges or isotropic texture—such as the “Cat’’ example (Figure 13)—show unstable or incorrect auxiliary peaks even when the main-task likelihood converges cleanly. These results confirm your intuition: classes with ambiguous or symmetric orientations indeed show lower—but still positive—correlations, while asymmetric classes consistently exhibit higher agreement between the two tasks. Despite such per-sample ambiguities, the auxiliary task remains stable at the batch level (appendix 7, figure 14), where its accuracy curves closely mirror the global trend of the main task’s inference dynamics.
>
> To quantitatively validate Assumptions A.1 and A.2, we now report Pearson and Spearman correlations (mean ± std over 3 seeds) for all 15 CIFAR10-C corruptions in the table above (Appendix 7, Table 4). We find that 11/15 corruptions display high positive correlations, supporting the assumptions at the batch level. The remaining four corruptions—Contrast, Defocus Blur, Motion Blur, and Zoom Blur—are exactly those that degrade directional cues, making rotation prediction unreliable. Even so, the mean accuracy gap across all corruptions remains only 0.984% (<1%), demonstrating that the auxiliary task still tracks the global inference dynamics well enough for effective peak-iteration selection.
>
> We kindly invite you to refer to the newly added visualization and analysis in Appendix 7, where these behaviors are shown explicitly.

---

> ### Author Response · Authors · 2025-11-18
>
> # Weakness 4, 5:
> - "In Lemma 1, how do we know that the task loss shrinks multiplicatively? Doesn't..."
> - "When looking at the curve comparing accuracy and norm difference (such as Fig 3),..."
> # Answer:
>
> Thank you for your thoughtful comments. We have taken your feedback into careful consideration and revised Section 5 to make the discussion clearer and more explicit. In addition, we have added Figure 8, which visualizes both the loss values and the loss variation across iterations, thereby strengthening the empirical support for our assumptions and clarifying the intuition behind our analysis. We have also revised Figure 3 in response to your comment regarding the comparison between accuracy–norm-difference curves and the onset of overthinking. Specifically, we standardized the x-axis ranges across relevant figures to make the overthinking region clearer (notice: we made a small adjustment to the decimal rounding of accuracy values, so it makes slightly shifting the reported peak-accuracy positions). We sincerely appreciate the reviewer for pointing out these aspects, which helped us improve both clarity and consistency in the visualizations.
>
> We respond below to your concerns regarding the interpretation of Lemma 1, the role of the dynamics, and how these relate to our discussion of overthinking. Specifically, we clarify what Lemma 1 is intended to show, how it connects to our empirical observations, and why our revision to Section 5 (along with the newly added Figure 8) makes the underlying intuition more explicit.
>
> ## (1) Clarifying the intention of Lemma 1 – we do not claim multiplicative shrinkage of the loss
>
> We would like to clarify that **Lemma 1 does *not* claim that the task loss shrinks multiplicatively or converges**. Instead, the lemma serves a different purpose.
>
> In prior work, *Bansal et al. (2022)* argue that divergence of the hidden state $\(h_t\)$ may lead to overthinking, and that convergence of $\(h_t\)$ may mitigate it. Our goal in Section 5 (“Discussion”) is to show that **convergence of $\(h_t\)$ alone is *not* the full picture**, and that overthinking depends on more nuanced dynamics than simply whether $\(h_t\)$ converges.
>
> ## (2) What Lemma 1 actually establishes
>
> Lemma 1 only shows the following:
>
> - The *change in the task loss between two consecutive states* is **upper-bounded by a monotonically increasing function** of $\(\|\Delta h_t\|\)$.
> - Therefore, **if** $\(\|\Delta h_t\|\)$ becomes very small (or converges to zero), **then the loss variation between consecutive states must also become very small**.
>
> This is a **bounded-drift result**, not a convergence or multiplicative-shrinkage result.
> It does **not** assume that the update is gradient-based, and therefore does **not** imply geometric decay of the loss.
>
> ## (3) Why small loss variation matters for overthinking
>
> Our working assumption in the Discussion is that:
>
> - After the model reaches **peak accuracy**, it should ideally maintain *stable* predictions,
> - Meaning: the loss variation between the peak state and its subsequent states should remain small.
>
> Therefore:
>
> - If $\(\Delta h_t\)$ is **small at the peak iteration**, then by Lemma 1 the **loss variation is also small**, helping the model avoid overthinking.
> - If $\(\Delta h_t\)$ is still **large at the peak iteration**, then the **loss variation remains large**, predictions fluctuate, and overthinking becomes likely.
>
> ## (4) Empirical evidence supporting this interpretation
>
> We revised Section 5 of the paper to make these arguments explicit and added **Figure 8**, which visualizes both the loss values and the loss variation across iterations.
>
> The key observations:
>
> - **Figure 3** shows that both Conv-GRU and Conv-LiGRU have convergent $\(h_t\)$, but **Conv-GRU still suffers from overthinking**, whereas Conv-LiGRU does not.
> - **Figure 8** shows that the **loss gaps** between states decrease in line with the behavior predicted by Lemma 1 (i.e., they shrink following $\(\|\Delta h_t\|\))$.
> - However, after peak accuracy:
>   - For **Conv-GRU**, the actual loss values begin to **diverge again**, consistent with the fact that at the peak iteration its $\(\Delta h_t\)$ is still large (~0.7).
>   - For **Conv-LiGRU**, the loss values **converge toward zero**, consistent with its very small $\(\Delta h_t\)$ at peak (~0.004).
>
> These observations together explain why Conv-GRU exhibits overthinking while Conv-LiGRU is more robust.
>
> ## (5) Contribution of our Discussion section
>
> From these analyses, we conclude that:
>
> - **Convergence of $\(h_t\)$ is not sufficient to avoid overthinking.**
> - The **magnitude** and the **rate of convergence** of $\(\Delta h_t\)$ also play a crucial role.
>
> Once again, we sincerely thank the reviewer for the constructive feedback, which has helped us improve the clarity and quality of our manuscript.

---

> ### Author Response · Authors · 2025-11-18
>
> # Weakness 6:
>
> The y axes for the two plots shown side by side in Figure 7 have very different scale: each unit means 10 for the plot on the left, and 5 for the plot on the right. For the argument about the unhalted model surpassing ACT on STL-10 to be more convincing, having similar scales will help.
>
> # Answer:
>
> Thank you for pointing this out. We agree that having different y-axis scales in the side-by-side plots of Figure 7 could make the comparison less straightforward. As suggested, we have redrawn Figure 7 using consistent y-axis scales across both subplots. While this adjustment does not materially change the visual trend, it provides a more uniform basis for comparing the behavior of the unhalted model and ACT on STL-10. The updated figure still reflects the qualitative observation we discuss in the paper—that allowing the model to think freely leads to performance that continues to improve beyond the very early stopping point chosen by ACT. We appreciate the reviewer for recommending this visualization refinement.

---

> ### Author Response · Authors · 2025-11-18
>
> # Weakness 3:
> - It is also not clear if the recurrent models are benefitting from extra compute at inference time, can you show a comparison that is FLOPs-matched?
>
> # Answer:
>
> Thank you for the question. In our setting, **“extra compute’’ corresponds directly to increasing the number of recurrent iterations at test time**. The purpose of our work is to analyze how accuracy changes as the inference steps increase—showing the initial gains and the eventual decline caused by overthinking. This trend is clearly visible in Figures 1, 3, 5, and 7.
>
> In this paper, we measure computation by the number of iterations, not by FLOPs. Each model has a fixed architecture and fixed number of parameters; the only quantity that changes during inference is how many recurrent updates the model performs. Since our goal is to study the effect of additional iterative computation itself, a FLOPs-matched comparison does not apply in this context.
>
> We will clarify this in the revised version to avoid ambiguity.

---

> ### Author Response · Authors · 2025-11-19
>
> # Weakness:
> - I am unclear on the details on training parity between the models: were BN, RandAugment, EMA, label smoothing etc used for training the feed-forward CNNs from scratch, was ViT trained with augmentations? These are important since these architectural changes are known to achieve higher CIFAR-C accuracy and show better OOD robustness. So, this would make the claim of Conv-GRU/Conv-LiGRU convincing.
>
> # Answer:
> Thank you for the helpful suggestion. As clarified in Section 4.2 and Appendix 6, all models in our study—including the feed-forward CNN, Conv-GRU, Conv-LiGRU, and the ViT baseline—are trained under strictly matched conditions: identical data augmentations, no normalization layers (no BN or GN following prior works (Schwarzschild et al., 2021; Veerabadran et al., 2023)), and the same self-supervised auxiliary task.
>
> We intentionally do not incorporate techniques such as RandAugment, EMA, or label smoothing. While these approaches can improve CIFAR-C robustness, they may also introduce effects that are independent of the iterative inference mechanism, making it harder to isolate and study the core behavior we focus on: how recurrent computation influences accuracy, including both its initial gains and its eventual overthinking dynamics.
>
> We appreciate the reviewer’s point, and examining how such robustness techniques interact with recurrent “deep thinking’’ models is indeed an interesting direction. We are happy to explore these extensions in additional experiments in future work.

---

> ### Author Response · Authors · 2025-11-20
>
> # Question:
>
> To test the scope of the OOD claim beyond narrow, artificially corrupted benchmarks such as CIFAR-C and Tiny-ImageNet-C that capture low-level pixel perturbations but not true domain or semantic shifts, what do you think about testing on benchmarks such WILDS, PACS etc? This would reveal whether the rotation-proxy criterion and iterative test-time scaling extend to real distribution shifts rather than only to synthetic corruptions.
>
> # Answer:
>
> Thank you for the valuable suggestion. We acknowledge that the term OOD in our initial framing was broader than the actual scope of the paper. We will revise the terminology to “distribution shift", which more accurately reflects the types of shifts we study.
>
> Regarding dataset choice, we intentionally use CIFAR-C and Tiny-ImageNet-C because their corruption levels provide a graded difficulty structure, similar to the difficulty scaling used in logical or algorithmic reasoning tasks. This allows us to examine a core aspect of deep-thinking models: harder inputs consistently require more recurrent iterations, while overly long inference leads to overthinking.
>
> In addition to these controlled distribution shifts, our paper also includes a cross-dataset shift via CIFAR-10 → STL-10, which is a separately collected dataset with higher resolution (96×96), different style, and different background statistics. As shown in Table 3, Conv-GRU outperforms the ResNet baseline on this real distribution shift, and our rotation-based estimator remains highly accurate, with only a 0.2% gap from the true peak and a strong Pearson correlation of 0.95.
>
> We agree that evaluating on larger benchmarks such as WILDS or PACS would further strengthen the work, and we will aim to run these experiments as soon as possible given our computational constraints.

---

### Official Review · Reviewer_j6qK · 2025-10-30

**Soundness:** 2
**Presentation:** 3
**Contribution:** 3
**Rating:** 4
**Confidence:** 3

**Summary:**

This paper studies deep thinking (recurrent models that improve by running more iterations at inference) in object recognition under OOD shifts. It identifies overthinking (accuracy drops after peak) and proposes a self-supervised proxy task (rotation prediction) to detect the optimal stopping point without test labels. The method is lightweight, avoids underthinking from prior halting rules, and shows strong gains on CIFAR-C, Tiny ImageNet-C, and STL10.

**Strengths:**

- The problem is interesting. Overthinking is real and under-addressed in vision OOD. Prior halting (ACT, norm-threshold) causes underthinking, limiting extrapolation.

- Using rotation prediction as a proxy is simple, label-free at test time, and doesn’t require parameter updates.

- Despite simple, the proposed method show strong empirical results in discussed setups.

**Weaknesses:**

- The method could fails if main and auxiliary tasks decouple under certain OOD shifts (e.g., texture, style, or adversarial). In practice, the models might need to deal with many OOD shifts after deployment. Additional ablation on alternative proxies or failure cases are needed.

- The corruption-based shifts is limited with only (CIFAR-C) and one domain shift (CIFAR→STL10). No semantic, style, or real-world OOD (e.g., ImageNet-R). No large-scale models (e.g., ViT-L, CLIP, DINO) or high-resolution inputs (224×224). This makes this paper as a proof-of-concept rather than a practical validation.

- The computational cost is not well discussed. Running 100 iterations per sample is slow in practice. There is a lack of latency analysis, early-exit trade-offs, per-sample iteration stats.

- The proxy must be trained jointly, not plug-and-play on pretrained models, limiting applicability to frozen backbones (e.g. foundation models). I wonder if we can do fine-tune the proxy task and main task from a pre-trained model rather than training them from scratch. By doing so, it would make this method more applicable since current computer vision research is largely based on pretrained backbones.

**Questions:**

See weaknesses

---

> ### Author Response · Authors · 2025-11-17
>
> # Weakness 1:
> The method could fails if main and auxiliary tasks decouple under certain OOD shifts (e.g., texture, style, or adversarial). In practice, the models might need to deal with many OOD shifts after deployment. Additional ablation on alternative proxies or failure cases are needed.
>
> # Answer:
> Thank you for raising the issue of potential decoupling between the main task and the auxiliary task. Since all three reviewers expressed similar concerns, we conducted an expanded set of analyses in the revised manuscript, presented in Appendix 7, covering both per-sample and batch-level behavior. These analyses are also discussed in detail in our responses to ***Question 1*** and ***Weakness 1*** from **Reviewer XTzY**, where we include the full Pearson/Spearman correlation table, per-sample visualizations, and batch-level accuracy curves.
>
> Therefore, we provide only a brief summary here: As detailed in Appendix 7, our per-sample analysis shows that examples with strong directional cues (e.g., “Ship’’) exhibit stable auxiliary rotation predictions that closely follow the main-task likelihood trajectory, whereas samples with weak or isotropic texture (e.g., “Cat’’) may produce unstable or incorrect auxiliary peaks, reflecting per-sample decoupling when orientation cues are ambiguous. However, at the batch level, the auxiliary task remains highly stable: its accuracy curves are smooth and consistent across classes and closely mirror the global inference dynamics of the main task. This is further supported by our full correlation study (Table 4), where 11 out of 15 CIFAR10-C corruptions show strong Pearson/Spearman correlations, while the four lower-correlated corruptions are precisely those that destroy directional structure. Even in these challenging cases, the average accuracy gap remains below 1%, indicating that the auxiliary task still reliably tracks the overall trend needed for peak-iteration selection.
>
> We kindly invite you to refer to Appendix 7 in the revised version, along with our detailed response to **Reviewer XTzY**, for the complete analysis.

---

> ### Author Response · Authors · 2025-11-20
>
> # Weakness:
> The computational cost is not well discussed. Running 100 iterations per sample is slow in practice. There is a lack of latency analysis, early-exit trade-offs, per-sample iteration stats.
>
> # Answer:
>
> Thank you for the comment. As shown in our correlation analysis (appendix 7 in the revision version) and the discussion in the comment above, the auxiliary and main tasks can diverge at the per-sample level for inputs with weak or isotropic structure, making per-sample early exit unreliable.
>
> For this reason, our method is not designed as a per-sample halting scheme. Instead, we intentionally run the model up to a fixed maximum number of iterations and use the auxiliary signal to estimate the optimal iteration at the batch level, where correlation is stable and reliable across corruptions.
>
> We acknowledge that the absence of per-sample halting is a current limitation of our approach, and developing reliable early-exit mechanisms for our self-supervised iteration estimation algorithm—despite sample-level variability—is a promising direction for future work.

---

### Author Response · Authors · 2025-11-20

# Global Comment (Large-Scale Experiments)

# Weakness/Question:

## Reviewer j6qK:
- The corruption-based shifts is limited with only (CIFAR-C) and one domain shift (CIFAR→STL10). No semantic, style, or real-world OOD (e.g., ImageNet-R). No large-scale models (e.g., ViT-L, CLIP, DINO) or high-resolution inputs (224×224). This makes this paper as a proof-of-concept rather than a practical validation.
- The proxy must be trained jointly, not plug-and-play on pretrained models, limiting applicability to frozen backbones (e.g. foundation models). I wonder if we can do fine-tune the proxy task and main task from a pre-trained model rather than training them from scratch. By doing so, it would make this method more applicable since current computer vision research is largely based on pretrained backbones.
## Reviewer pdNM:
- needs more results on other datasets: the paper reports results on cifar10. however, following hendryks et al, and existing literature in test-time-training, results on bigger datasets like imagenet-v2, imagenetsketch, imagenet-a, imagenet, and 10 fine-grained datasets should be reported.

# Answer:

We sincerely thank all reviewers for highlighting the importance of validating our findings on larger-scale datasets and models. We fully agree that extending the experiments beyond CIFAR-C and STL10—e.g., to higher-resolution inputs, stronger pretrained backbones, or more complex vision tasks—would further strengthen the contributions of this work.

Due to limited computational resources (single RTX 3090), we were unable to run full ImageNet-scale experiments during the initial submission. However, in response to the reviewers’ requests, we are actively running additional large-model and high-resolution experiments during the discussion period. In particular, we are evaluating a pretrained Mask R-CNN + FPN system where the Conv-Upsample head is replaced by a Conv-GRU recurrent head, and reporting AP across different recurrent iteration counts. These experiments more closely reflect real-world vision pipelines and pretrained model usage.

We are doing our best to complete and report these results before the end of the rebuttal window, and we kindly ask the reviewers for understanding as these evaluations may take time to finish on our limited hardware.

We appreciate the reviewers’ constructive suggestions and view these larger-scale extensions as an important next step for the broader applicability of our method.

---

### Meta-Review · Area_Chair_aoQM · 2025-12-04

**Summary:**

The key reviewer concerns that I identified were:

1. A lack of large scale experiments
2. Concerns regarding novelty

As outlined below, I do not think 1 was addressed (and 2 was perhaps partly addressed). All scores are negative, so I propose rejection. I encourage the authors to include larger scale experiments for their next submission.

**Reviewer Concerns:**

Multiple reviewers were concerned at the lack of larger scale experiments. Unfortunately, the authors weren’t able to resolve this (the submission has not been updated since 20th November). I think the authors have made a good attempt at defending the novelty of their method, although it is not fully convincing (although that said, this is not as important an issue as the lack of experiments from my reading).

**Reviewer Scores:**

I think there is a possibility Reviewer XTzY would have raised their score (perhaps 2->4) with the correlations provided. I do not think any reviewers would have raised their scores without Imagenet (or similar) experiments, although for what it’s worth the author-reviewer engagement with Reviewer pdNM is commendable.

---

### Decision · Program_Chairs · 2026-01-26

Reject